# Cell of origin epigenetic priming determines susceptibility to *Tet2* mutation

Giulia Schiroli [1,2,3], Vinay Kartha[2,3,4,5], Fabiana M. Duarte [2,3,4,5], Trine A. Kristiansen[1,2,3], Christina Mayerhofer[1,2,3], Rojesh Shrestha[2,3,4], Andrew Earl [2,3,4], Yan Hu[2,3,4], Tristan Tay[2,3,4], Catherine Rhee[1,2,3], Jason D. Buenrostro [2,3,4,6] ✉ & David T. Scadden [1,2,3,6] ✉

Hematopoietic stem cell (HSC) mutations can result in clonal hematopoiesis (CH) with heterogeneous clinical outcomes. Here, we investigate how the cell state preceding *Tet2* mutation impacts the pre-malignant phenotype. Using an inducible system for clonal analysis of myeloid progenitors, we find that the epigenetic features of clones at similar differentiation status are highly heterogeneous and functionally respond differently to *Tet2* mutation. Cell differentiation stage also influences *Tet2* mutation response indicating that the cell of origin's epigenome modulates clone-specific behaviors in CH. Molecular features associated with higher risk outcomes include *Sox4* that sensitizes cells to *Tet2* inactivation, inducing dedifferentiation, altered metabolism and increasing the in vivo clonal output of mutant cells, as confirmed in primary GMP and HSC models. Our findings validate the hypothesis that epigenetic features can predispose specific clones for dominance, explaining why identical genetic mutations can result in different phenotypes.

Upon aging, somatic cells steadily accumulate mutations at a rate of approximately 10–50 novel mutations per year across various tissues[1–6]. Every organ can be defined as a mosaic of cell clones carrying distinct mutations. Although most of these mutations are of little or no functional consequence, some may confer fitness advantage. A well-studied example can be found in the hematopoietic system, which is maintained by a pool of self-renewing multipotent hematopoietic stem cells (HSC). Upon aging, the long lifespan of HSC makes them particularly susceptible to the accumulation of genetic mutations; best estimates suggest that HSC acquire 1-3 exonic somatic mutation per cell per decade[7,8]. Among these, mutations in epigenetic regulators are commonly found in healthy elderly individuals with normal blood counts in a process called Clonal Hematopoiesis (CH), leading to an increased clonal fitness and positive selection over time[9]. CH is associated with an increased risk of all-cause mortality, a >10-fold increased risk of hematological malignancies, cardiac disease development and

overall clinical adverse outcome[5,10–13], representing a precursor state for myelodysplastic syndromes (MDS) and ultimately acute myeloid leukemia (AML)[14–16].

One poorly understood aspect of CH is its incomplete penetrance, with most (~96%) patients remaining cancer-free over time, suggesting additional alterations to cells are needed for cell transformation. Current approaches for risk prediction are based on retrospective examination of the genetic mutations[9,17,18]. Clone size, driver gene identity, number of mutations and the fitness rate of specific variants have been demonstrated to have predictive power for transformation risk. However, while driver mutations provide useful insights into clonal progression, recent reports demonstrated that they are not complete indicators of risk. Indeed, high heterogeneity has been reported in the growth rate of clones harboring the same genetic mutation among different patients[19] and a large proportion of CH cannot be linked to known driver genes[20], suggesting that a large

[1]Center for Regenerative Medicine, Massachusetts General Hospital, Boston, MA 02114, USA. [2]Department of Stem Cell and Regenerative Biology, Harvard University, Cambridge, MA 02138, USA. [3]Harvard Stem Cell Institute, Cambridge, MA 02138, USA. [4]Gene Regulation Observatory, Broad Institute of MIT and Harvard, Cambridge, MA 02142, USA. [5]These authors contributed equally: Vinay Kartha, Fabiana M. Duarte. [6]These authors jointly supervised this work: Jason D. Buenrostro, David T. Scadden. ✉e-mail: jason_buenrostro@harvard.edu; david_scadden@harvard.edu

fraction of the unexplained variability can be ascribed to other cell extrinsic or intrinsic factors[21]. Environmental stimuli, such as changes in the bone marrow (BM) environment[22], inflammation[23] or transplant-related proliferative stress[24] can influence clone growth. Alternatively, we hypothesize that the rate of clonal expansion can be modulated by specific properties of the clone of origin. While long-term consequences are mediated by HSC, downstream phenotypic effects such as myeloproliferation can als be influenced at the progenitor level.

Using clonal tracing methods, we and others previously found that functional heterogeneity[25] among unmutated hematopoietic clones is associated in vivo with defined molecular landscapes, both at the epigenetic[26] and transcriptomic level[27]. In accordance, clonal heterogeneity is also emerging as an important cause of incomplete penetrance of leukemic mutations[28]. In previous work, we demonstrated that clonal behaviors were largely scripted in the epigenome of the cell of origin[26]. These clone-specific functional differences include responses to inflammatory and genotoxic stress[26]. It is likely, therefore, that the response to a genetic mutation will similarly be influenced by the epigenome of the clone of origin.

As a model to demonstrate this concept, we investigated the epigenomic dependencies of Tet methycytosine dioxygenase 2, *Tet2*. Inactivating mutations to *TET2* are amongst the most prevalent in CH and are associated with the highest relative risk of disease development[14]. Mutations to this epigenetic regulator are associated with hypermethylation of cytosines at enhancer sequences and alterations in physiologic hematopoietic differentiation[29]. Knock-out (KO) mouse models accurately mirror human disease progression, with a fraction of mice transitioning from CH to different hematologic disorders months after the induction of the mutation[30,31]. Several studies have suggested that mutations to *Tet2* can have diverse functional roles across disease relevant cell types. In HSCs, *Tet2* mutations skew the HSC transcriptome towards a myelo-monocytic fate[32] and expand the hematopoietic progenitor cell population in a cell-intrinsic manner[30], resulting in altered differentiation and function particularly of myeloid populations[33].

In this study, we characterize the epigenetic and functional consequences of *Tet2* inactivating mutation in early, committed and differentiated primary mouse BM populations. By pairing this information with an ex vivo clonal system based on Granulocytic-Monocytic Progenitors (GMP), we uncover a clone-specific epigenetic state which affects the extent of functional changes seen in myeloid cells. Further, by using HSC transplantation models we highlight that this "sensitizing" epigenetic state coincides with in vivo outcomes in mice, increasing the molecular and functional dysregulation associated with *Tet2* mutation.

Collectively, these results provide experimental evidence that epigenetic features of the cell of origin influences the phenotype of *Tet2* mutant clonal hematopoiesis.

## Results

### Characterization of the *Cis*-regulatory landscape of GMPs in vivo

To study how inactivation of an epigenetic regulator may promote disease-associated programs, we performed molecular profiling at the single cell level of bone marrow (BM) hematopoietic progenitor cells from inducible *Tet2* mutant mice (Mx1-Cre *Tet2*fl/fl[30]) or matched controls from the same genetic background (*Tet2*fl/fl, referenced as WT-wild-type) (Fig. 1A). All the mice were treated with polyinosinic-polycytidylic acid (pI:pC) to account for inflammation-induced alteration of cell behavior and phenotype[34]. Efficient *Tet2* deletion was confirmed at the genomic level by ddPCR in >95% of circulating cells (Supplementary Fig. 1a). Upon *Tet2* deletion, phenotypic analysis highlighted expansion of myeloid cells in the periphery, splenomegaly in a fraction of mice, and accumulation of primitive HSC and myeloid progenitors in the BM, consistent with previous reports for this model (Supplementary Fig. 1b-e)[30].

Our single-cell dataset (n = 4 *Tet2* KO mice, n = 2 WT mice) included populations spanning from HSC to mature myeloid effector cells (Lin- progenitors, CD11b+ cells), and sorted GMPs (Lin- cKit+ Sca1- CD34 + CD16/32 + ), thus encompassing all the different intermediates of myeloid differentiation (Supplementary Fig. 1f, g, Supplementary Data 1). GMPs represent a heterogeneous mixture of myeloid-restricted cells which connect primitive progenitors to committed multi-lineage myeloid fates[35,36]. This latter cell type is particularly relevant during oncogenic transformation where MDS/AML cells progressively co-opt GMP identities[37,38].

We analyzed a total of 57,232 cells for scATAC-seq[39] and 8,969 cells for scRNAseq and applied our previously developed analytical framework to link chromatin accessibility and gene expression across cells[40]. In this way, we were able to annotate the scATAC-seq data using known markers for hematopoietic BM populations[35,36] extracted from the paired single cell RNA-seq dataset (Fig. 1b, Supplementary Fig. 1h, i). Expression of representative genes for primitive or committed populations is overlaid on a 2D UMAP projection of single cells (Fig. 1c).

We focused on defining the epigenetic heterogeneity that underlies GMP identity. We adopted an unbiased approach and clustered GMP based on accessibility profile for the 50 most variable transcription factor (TF) motifs after removing contaminant populations from the sorting procedure (Fig. 1d, top panel). This analysis revealed a high degree of complexity within GMPs, suggesting the presence of multiple chromatin states. Accessibility of some key TF correlates with distinct myeloid cell identities (for example, Gata, Tal1 motifs for stem-like cells, Cebpe for granulocytic or Irf motifs for monocytic differentiation). Interestingly, high accessibility for TFs like Tcf3/4 and Runx1/2 identified states of transition between primitive GMP and more committed states consistent with a role of these factors in regulating differentiation[36,41]. Other TFs like Nfkb, Jun, Fos, Nfia/x showed heterogeneous distribution patterns across all the different myeloid states and likely reflect activation of other regulatory properties such as inflammation and cell cycle.

To link chromatin accessibility features to downstream gene expression, we identified domains of regulatory chromatin (DORCs), which prime cell identity for activation of gene expression as described during epithelial lineage commitment[40]. These regions are defined by first ranking genes based on their total number of significantly correlated peaks and focusing on genes with ≥ 3 associated accessibility peaks (n = 3474, Supplementary Fig. 2a, b). DORC-associated genes identified here for hematopoietic progenitors included many known mediators of lineage specification and differentiation[35] such as *Gata1* and *Mpo*. Direct comparison between differential DORCs and cognate RNA expression revealed a general correlation across cells (Fig. 1d, middle and bottom panel) and highlighted the clear presence of 3 defined GMP states (stem-like, neutrophilic and monocytic). These data suggest that enhancer-like chromatin features are able to predict differentiation trajectories and cell identity during hematopoietic differentiation and suggest a differentiation model where GMP commitment starts from early progenitors which co-express stem and myeloid-related genes, and then markedly bifurcate into monocytic or granulocytic fates. Interestingly, for subsets of DORCs, we observed a gain in accessibility prior to the onset of their associated gene's expression[40]. For example, for Klf4, a key TF determinant of the monocytic lineage, we detected DORC activation prior to gene expression and before lineage commitment; these are quantified as 'residuals' (difference of chromatin accessibility and expression of the gene[40]) (Supplementary Fig. 2c). These data indicate that chromatin accessibility can foreshadow gene expression and may offer an advantage over transcriptomic studies in capturing alterations in cell state and functions in downstream lineage. In particular, it can be used to capture subtle differences in cell states introduced by *Tet2* KO (Supplementary Fig. 2d).

As a parallel analysis, we defined the core regulatory epigenetic program of GMPs by aggregating accessible chromatin regions co-regulated by related sets of TFs[42]. In this manner, we identified 3 distinct signatures (co-accessibility modules, see methods, Supplementary Data 2) that represent relevant myeloid functional states, as shown by plotting module accessibility on UMAP (Fig. 1e) or by looking at the TF motifs and gene scores correlated to each peak module (Supplementary Fig. 2e). A development-related signature was indeed associated with GMP-stem identity, as shown by a gain in Runx1/2, Gata and Tal motif accessibility, whereas GMP-neutrophilic primed signature

included gain of accessibility in the Cebpe motif and genes implicated in neutrophil differentiation (*Mpo, Ctsg, Elane*). The signature for GMP-monocytic priming was associated with enrichment in Irf, Klf motifs and *Spi1, Ccl2, Ly86* gene scores. Accordingly, pathway enrichment analysis using top correlated genes to each signature confirmed significant association with expected functional myeloid identities (Supplementary Fig. 2f).

Finally, we analyzed how *Tet2* mutant cells are localized across the differentiation trajectory. In accordance with results from previous RNA-seq studies[32], we observed that *Tet2* knockout does not result in

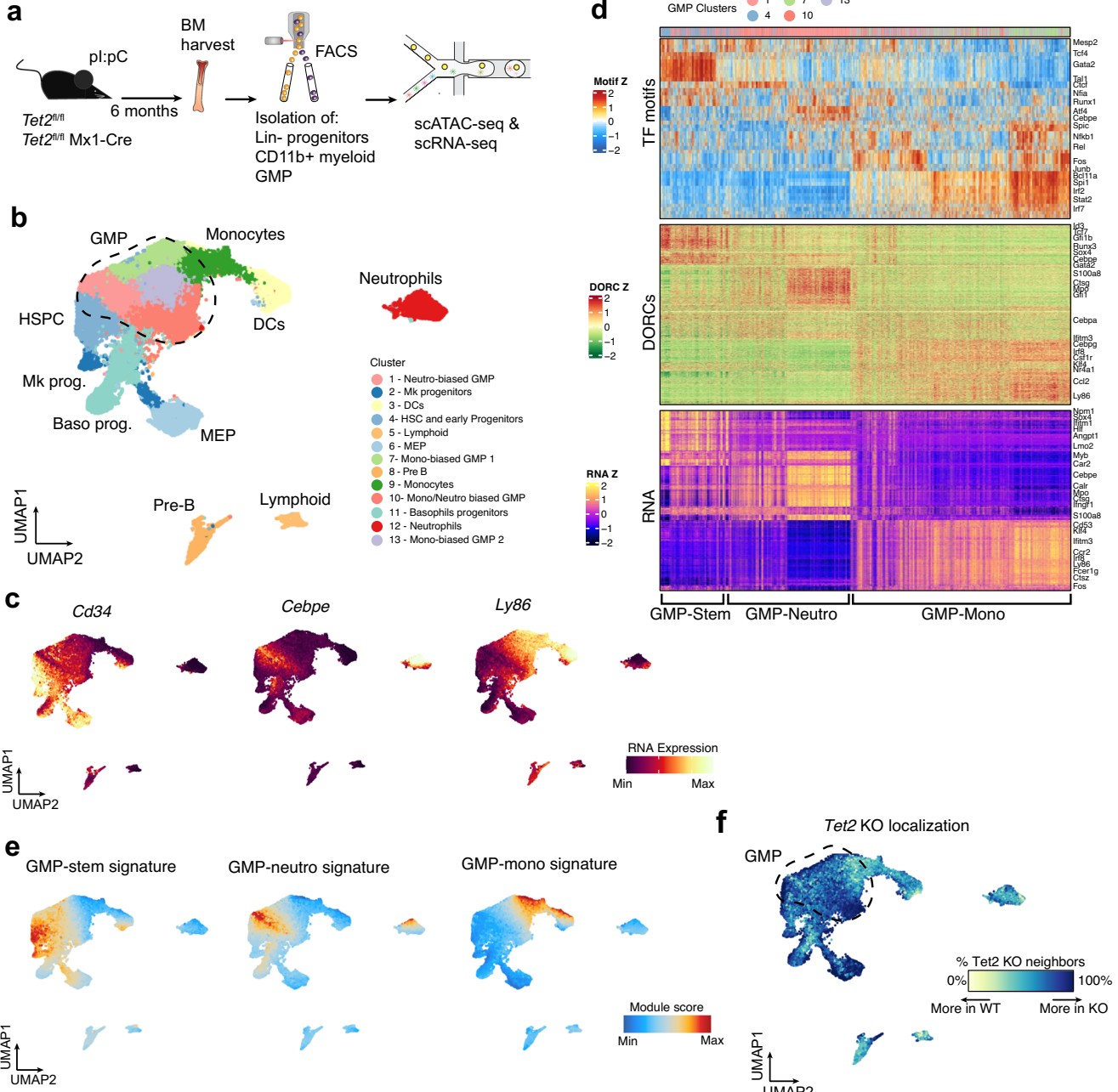

**Fig. 1 | Definition of chromatin states that describe in vivo GMP differentiation.** **a** Cartoon depicting experimental strategy and molecular analyses for primary *Tet2* KO and WT populations. **b** UMAP plot showing the combined scATACseq coordinates and annotation of sorted Lin- progenitors, CD11b+ myeloid cells and GMPs. HSPC Hematopoietic Stem and early Progenitor Cells, GMP Granulocytic-Monocytic Progenitors MEP Megakaryocyte-erythroid progenitors, DCs Dendritic Cells. **c** UMAP plots showing RNA expression of representative markers for stem and early progenitors (left), granulocytic (middle) or monocytic (right) differentiation. **d** Heatmaps comparing TF accessibility (top), DORC accessibility (middle) and RNA expression (bottom) of most variable elements across GMP cells. Cells are color coded by cluster identity, as indicated. **e** UMAP plots showing cumulative peaks accessibility for GMP chromatin signatures. **f** Plot showing relative distribution of WT and *Tet2* KO cells across the differentiation trajectory. Source data are provided as a Source Data file.

independent clustering (Supplementary Fig. 1g), but rather in the redistribution of GMP cells within selected neighborhoods (Fig. 1f). This suggests that *Tet2* mutation perturbs the chromatin state of primitive hematopoietic progenitors, but effects on cell identity are minimal since no significant skewing in cell abundances across clusters was observed (Supplementary Fig. 1j, k).

Taken together, our unbiased analysis defined epigenetic programs relevant for defining functional identities of primary GMP cells, which are largely consistent with hierarchical differentiation models previously defined by transcriptomic analyses[43,44]. These GMP signatures represent relevant epigenetic functional states and serve as a blueprint to further map *Tet2*-mediated distortion of GMP states.

### *Tet2* is required for correct priming and function of myeloid progenitors

To investigate how Tet2 deficiency influences myeloid cell identity at a molecular level, we analyzed differential representations within the scATAC seq and scRNA seq datasets. Differential chromatin accessibility analysis was performed in the bulk GMP population (correcting for cluster ID, see methods) to increase statistical power (Supplementary Fig. 3a) and demonstrated 2,485 peaks upregulated and 2,621 peaks significantly downregulated upon *Tet2* KO (FDR ≤ 0.01) (Fig. 2a, Supplementary Fig. 3b). To confirm that these chromatin changes are related to direct binding of Tet2, we overlapped the differential peaks with available *Tet2* ChIP-seq data from myeloid progenitors immortalized by AML1-ETO[45]. We indeed found overlap for a high fraction of differential accessibility peaks (Fisher exact-test $P < 0.001$), finding 30.9% for peaks repressed and 21.8% for peaks induced upon *Tet2* KO. In accordance with the *Tet2* function as a DNA demethylase[29], a larger fraction of accessible regions predicted to be bound by Tet2 are observed to be repressed upon *Tet2* KO. In order to evaluate the level of epigenetic dysregulation on a single cell basis, we calculated single cell scores for accessibility of *Tet2* differential peaks from Fig. 2a (Fig. 2b). We observed a high spread in the distribution of *Tet2* KO cells as compared to WT (especially for peaks upregulated in *Tet2* KO), suggesting that there is a level of heterogeneity in how single GMP cells are epigenetically affected by *Tet2* mutation.

We then further investigated the functional relevance of the chromatin regions differentially regulated after *Tet2* knockout. To do this, we calculated enrichment of TF motifs among differentially accessible chromatin peaks and used these as an indicator of potential TF activity. We detected significant enrichment of many TF families within differentially regulated peaks comparing WT and *Tet2* KO GMP cells (permutation $Z$-test FDR < 0.001; Fig. 2c). Specifically, among the activated peaks, we observed a strong enrichment of Runx1/2 motifs: TFs involved in cell differentiation at early progenitor stages. The repressed peaks, in contrast, were enriched for Bcl11a, Irf8, Stat, Atf4 motifs: TFs that regulate monocytes and granulocytes cell commitment and differentiation. Interestingly, across both upregulated and downregulated peaks inflammation related TF motifs (Nfkb for activated, Irf1/8 for repressed) were identified, consistent with an enhanced inflammatory state as previously observed in *Tet2* KO models[46,47]. These alterations collectively highlight an impairment of myeloid differentiation in response to *Tet2* knockout. We also consistently observed a decrease in Ctcf motifs, which are sensitive to DNA methylation (Fig. 2c). This finding correlates with reduced CTCF binding previously observed in IDH mutant gliomas[48,49], and might indicate overall disruption of chromatin insulation in *Tet2* KO in the hematopoietic context.

As a complementary approach, we assessed the differentiation stage where *Tet2*-differential chromatin regions calculated in GMPs are activated (Fig. 2d). Peaks induced upon *Tet2* KO showed high accessibility in primitive cells, whereas peaks repressed upon *Tet2* KO showed highest accessibility upon progenitor commitment towards myeloid fates. These data are consistent with *Tet2* being required for

priming of GMPs towards both monocytic and granulocytic lineages. To validate these results in an orthogonal way, we analyzed our defined epigenetic signatures for GMP cells from Fig. 1 comparing WT and *Tet2* KO cells (Fig. 2e). Upon *Tet2* KO, we observed significant loss in accessibility for the GMP-monocytic signature and an increase in accessibility in the GMP-stem signature, further indicating that chromatin changes mediated by Tet2 alter the core regulatory network at different levels of myeloid differentiation.

Furthermore, differential testing among DORCs identified extensive alterations at the level of regulators of myeloid differentiation across different GMP subclusters (Supplementary Fig. 3c, Supplementary Data 3).

Since our epigenetic analysis indicates extensive chromatin remodeling, we analyzed the downstream alterations at the level of gene expression. *Tet2* is expressed throughout the hematopoietic differentiation hierarchy, although poorly detected in our scRNAseq dataset (Supplementary Fig. 3d). Enrichment analysis comparing *Tet2* KO and WT cells was conducted for cell subsets covering the key stages of myeloid differentiation (Fig. 2f). Among the upregulated pathways, gene sets were identified that broadly associate with altered oxidative state, translation, response to cell pathogens and inflammation across the granulocytic and monocytic cell fates. Among the most significant terms for HSPC (Hematopoietic Stem and early Progenitor Cells) we found MHC class II antigen presentation, in accordance with this recently described immunosurveillance mechanism being used in safeguarding pre-malignant states[50]. Downregulated terms indicate a differentiation arrest of GMP as well as substantial alterations in physiologic myeloid activation and functions in the absence of *Tet2*.

Finally, to validate whether the priming defect we observed in GMP cells upon *Tet2* KO leads to functional alterations after differentiation, we analyzed the cell functionality of monocytes and neutrophils with inflammatory stimuli. Mature *Tet2* KO myeloid cells from both lineages showed increased reactive oxygen species (ROS) production upon treatment with PMA but not in basal conditions (Fig. 2g). We also observed decreased phagocytosis in *Tet2* KO cells (Fig. 2g), in line with functional alterations described for Tet2-deficient monocytes and neutrophils[51,52].

Overall, these data shed light on the *Tet2*-related molecular mechanisms involved in correct myeloid cell priming that influence downstream effector functions.

### Linking epigenetic states to functional properties of GMP clones

We next sought to determine how diverse GMP cell states may be differentially altered in their function and in response to *Tet2* knockout. We hypothesized that individual GMP clones may stably inherit epigenetic, transcriptomic and functional states. To perform paired functional and molecular analyses on single hematopoietic clones, we took advantage of a high-throughput in vitro model of primary myeloid progenitors conditionally immortalized using Hoxb8 retrovirus fused to Estrogen Receptor domain (Hoxb8-ER)[53]. This culture system enables extended self-renewal of primary GMPs, allowing their single-cell isolation and indefinite expansion as pure clones. Upon withdrawal of the estrogen analog, the clones undergo physiological differentiation into mature myeloid cells with intact molecular characteristics and in vivo functions of primary cells[54]. We generated a dataset of 28 individual clones from a *Tet2*fl/fl mouse and each of them was divided into cohorts for induction of *Tet2* KO or control (Fig. 3a). Each clone pair is derived from an individual GMP cell. Transient Cre induction was obtained by transducing the cells with Cre-encoding integrase-defective lentiviral vector (or GFP as control), resulting in nearly complete introduction of *Tet2* targeted deletion in all the clones, as measured by a quantitative PCR assay (mean 97%, Supplementary Fig. 4a). 24 clone pairs were generated in the presence of SCF, which primes them towards granulocytic fate, and 4 of them were derived in

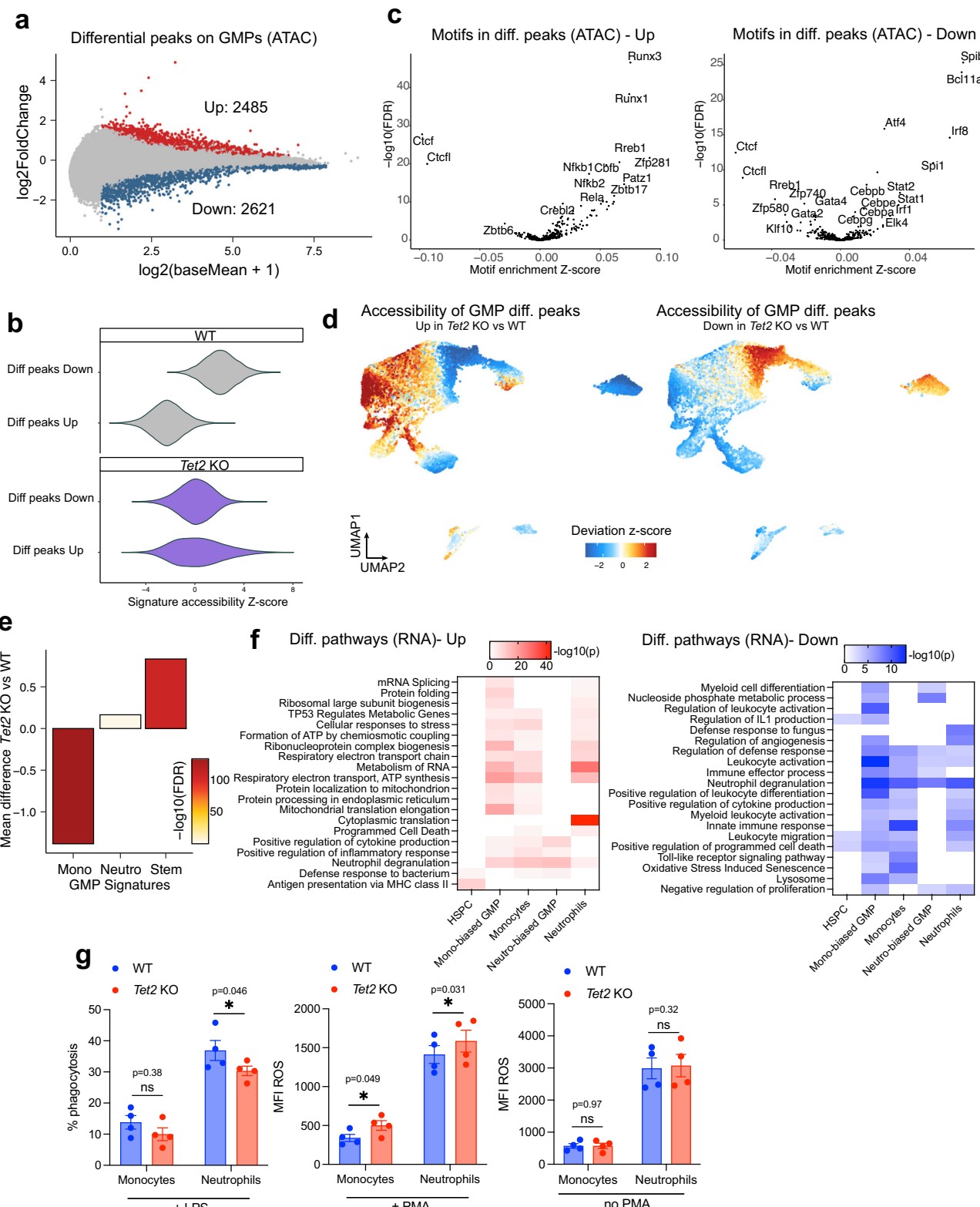

the presence of GM-CSF, which arrests them at a more mature stage and directs them towards a monocyte-macrophage fate (Supplementary Fig. 4b-d). We profiled the epigenetic landscape of the clones by performing multiplexed bulk ATAC-seq. Our analyses revealed highly heterogeneous patterns of chromatin accessibility (Supplementary Fig. 4e, f), as expected from previous studies[55], with separate clustering of SCF and GM-CSF clones (Fig. 3B, left panel). We also observed some genotype-specific clustering, suggesting that *Tet2* KO

extensively alters the chromatin accessibility profile of GMP clones (Fig. 3b, right panel).

We then characterized cell function by performing multiple in vitro functional assays on the differentiated progeny derived from clonal progenitors (Fig. 3c, Supplementary Fig. 5a–d). We measured the expression of myeloid markers, production of inflammatory cytokines (IL6), cell competency to clear pathogens in basal and stimulated conditions by ROS generation and phagocytic ability.

**Fig. 2 | *Tet2* is required for correct myeloid cell priming. a** Plot showing the fold change in the accessibility of chromatin peaks comparing WT and *Tet2* KO GMP cells. Cells belonging to clusters 1,4,7,10,13 for GMP samples were included. Significantly upregulated peaks (FDR < 0.01) are colored in red while significantly downregulated peaks are colored in blue. **b** Z-score accessibility of differential peaks in GMPs from Fig. 2a measured per-cell relative to the entire dataset. Genotype is indicated. **c** TF motif accessibility comparing WT and *Tet2* KO GMP cells measured in differential peaks regions (Left: upregulated, Right: downregulated). **d** Cumulative accessibility of differential upregulated (left) and downregulated (right) GMP peaks from Fig. 2a plotted on UMAP coordinates. **e** Plot showing the mean difference in GMP chromatin signatures from Fig. 1 comparing WT and *Tet2*

KO cells. Level of significance is color coded. **f** Pathway enrichment analysis (performed using Metascape[115]) of upregulated (left) and downregulated (right) transcripts in the indicated cell populations. Significance level is color coded. *p*-values are calculated based on the two-sided cumulative hypergeometric distribution, no adjustment for multiple comparisons is reported. **g** Functional assessment of WT and *Tet2* KO primary monocytes and neutrophils, showing % phagocytosis with LPS stimulation (left), production of ROS with PMA stimulation (middle), production of ROS without PMA stimulation (right) *n* = 4 independent mice, two-tailed paired *t*-test; exact *p*-values are indicated on the figure. Data are presented as mean values +/- SEM. Genotype is color coded. Source data are provided as a Source Data file.

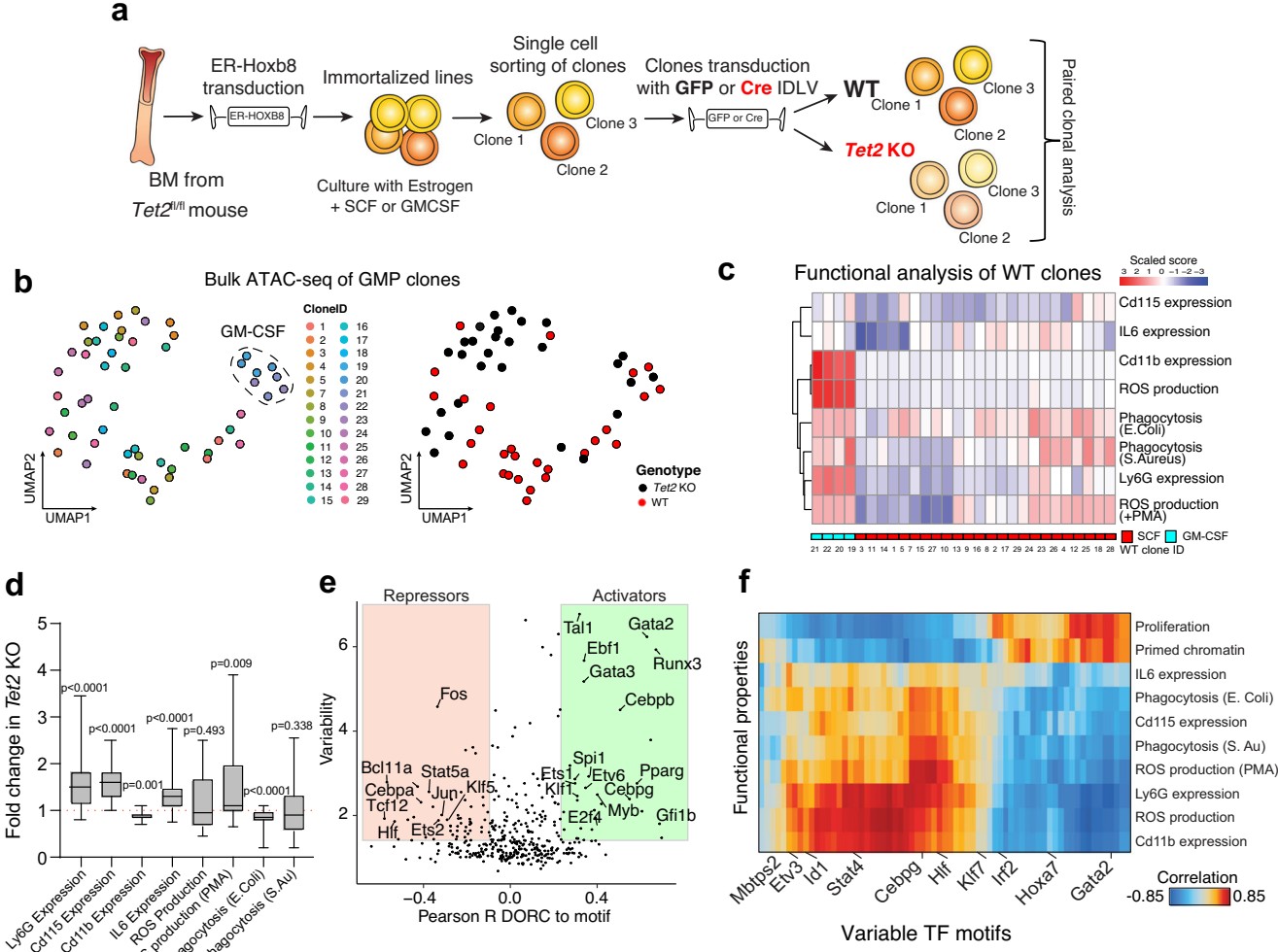

**Fig. 3 | GMP system as a model for heterogeneity reveals clone-specific changes. a** Cartoon describing experimental workflow for the generation of paired WT and *Tet2* KO single GMP clones using Hoxb8-ER system and relative analyses. IDLV Integrase-Defective Lentiviral Vector. **b** UMAP plot showing heterogeneity in the chromatin profile of different clones by high-throughput bulk ATAC-seq analysis. Color coding by clone ID and cell type is shown on the left (dashed line: clones generated using GM-CSF; the rest of the clones were generated with SCF). Color coding by genotype is shown on the right plot. **c** Heatmap showing performance of different WT GMP clones assayed for the indicated in vitro functional assays. GM-CSF and SCF clones are color-coded. **d** Paired analysis showing fold change

difference between WT and *Tet2* KO clones tested in assays from Fig. 3c *n* = 28 clone pairs, two-tailed Wilcoxon matched-pairs signed rank test. Exact *p*-values are indicated on the figure. Box: 25th to 75th percentile; line: median; whiskers: min to max. **e** Plot showing Pearson correlation between chromatin accessibility of DORC and respective TF motifs in GMP clones. Positive correlation (green) indicates activators whereas negative correlation (red) indicates repressors. Both GM-CSF and SCF clones were included in this analysis. **f** Heatmap showing Pearson correlation between TF activity measured by motif accessibility and cell function of GMP clones measured by the indicated functional assays. Both GM-CSF and SCF clones were included in this analysis. Source data are provided as a Source Data file.

Unsupervised clustering revealed the presence of distinctive functional groups among WT clones (Fig. 3c), supporting Hoxb8-ER as a powerful model for investigating myeloid clonal heterogeneity. Overall, GM-CSF clones showed increased potency in effector functions, in accordance with their higher basal differentiation state. SCF

clones showed high heterogeneity of behavior, with some clones showing potent effector functions while others were substantially less proficient. Moreover, chromatin tracks in GMP cells showed correlation with effector function upon differentiation (Supplementary Fig. 5e).

We then evaluated functional alterations mediated by *Tet2* KO (Fig. 3d, Supplementary Fig. 5a–d). There was significant alteration in myeloid differentiation marker expression, reduction of phagocytosis of *E.Coli* bacterial particles, increased production of ROS and production of IL6 upon PMA and LPS stimuli. The alterations collectively indicate hyperresponsiveness to inflammatory stimuli and impaired host anti-bacterial defense. Importantly, these results mirror functional changes observed in primary *Tet2* KO cells in vivo by previous reports[47,56,57] and in our own primary dataset of primary monocytes and neutrophils (Fig. 2g).

We next sought to determine whether these functional alterations may be explained by heritability of chromatin states at the progenitor level. We collectively analyzed WT SCF-derived and GM-CSF derived clones to capture major components of heterogeneity. We correlated accessibility of variable TF motifs with those of their cognate DORCs. This analysis identified activators and repressors which regulate chromatin accessibility including many known TFs involved in differentiation (Runx1, Hlf and Tal1), GMP specification (Cebpa-b), effectors of myeloid cell function, (Nr4a1-3 and members of the Atf family[58,59]) (Fig. 3e). Importantly, performing correlation analyses between the functional data and significant TF regulators, we could directly link the epigenetic states of the GMP progenitors to their proficiency in performing effector functions (Fig. 3f). We observed high correlation with proliferative potential and in vivo output capacity with stem-associated transcription factors (members of the Runx, Gata, Hoxa families, Sox4 and Tal1). Most effector functions correlated with Nr4a1, Atf, Cepb and Stat family members. IL6 and Cd115 expression mostly correlated with Nfkb2 and Klf5, consistent with their described role in inflammatory responses[60,61].

Collectively, these data show that the GMP clonal system is a robust model that enables the study of myeloid heterogeneity and directly links epigenetic features with relevant cell functions, which we found altered with *Tet2* KO. It further highlights that effector function upon differentiation of single clones can be related to specific features in the chromatin accessibility profile of the progenitor of origin.

### Paired clonal analysis defines clone-specific sensitization to *Tet2* mutations

We then investigated whether different GMP clones have different levels of molecular perturbation after *Tet2* KO. The identification of molecular determinants of sensitivity to the mutation might inform which clones will be more affected in their function (and thus likely to drive myeloproliferation/disease development). We thus compared functional alterations within WT and mutated clone pairs (Fig. 4a). Unbiased clustering showed that some clones had a significantly altered phenotype in response to *Tet2* KO while others were largely unperturbed in their functions. We then summarized the extent of functional perturbation of each pair of sister WT and *Tet2* KO clones by calculating a "Functional Perturbation Score", computed as the euclidean distance of each clone pair when considering all functional in vitro data (Fig. 4a, see methods). Here, comparison between SCF and GM-CSF groups allows evaluating the effect of *Tet2* KO between different differentiation states, whereas comparison within SCF clones ($n = 24$) allows comparison of clonal states within the same cell type. We found that GM-CSF derived clones were less perturbed in their behavior compared to SCF derived clones (Fig. 4b, GM-CSF vs SCF), possibly because these cells are immortalized at a differentiated stage whereas our in vivo data suggest *Tet2* is mostly required earlier in the hematopoietic hierarchy. In accordance, high perturbation correlated with lower differentiation properties (Supplementary Fig. 6a). Focusing on the heterogeneity among SCF clones (Fig. 4b), some clones showed a higher extent of functional alteration after *Tet2* KO. As an additional functional parameter of self-renewal ability, we tested the in vivo output of the SCF GMP clones by transplantation in lethally irradiated recipients. GMP clones differentiate in vivo and give rise to a transient wave of granulocytic progeny in the peripheral blood (PB) in accordance with their limited self-renewal potential (Supplementary Fig. 6b). Notably, in vivo output of the transplanted clones was collectively higher for *Tet2* mutants and we observed again clone-specific divergent responses (Fig. 4c). Collectively, these data suggest that Tet2 impacts in vitro and in vivo functions of myeloid progenitors based on both the level of differentiation and the specific clonal features within a given differentiation state.

Since we have shown that the GMP system is suitable to directly correlate epigenetic state and cell function, and that different clones are functionally impacted by *Tet2* KO to different extents, we investigated whether there are epigenetic states in the cell of origin which affect the response to *Tet2* mutation. We set up an analytical framework to predict sensitivity to *Tet2* KO by correlating functional responses in paired KO and WT clones and differential epi-signatures defined in our in vivo primary dataset (Fig. 4d). We first ranked WT clones by Functional Perturbation Score and assessed correlations of accessibility of chromatin loci with regulatory activity (DORCs) in primary cells. This genome-wide unbiased analysis defined DORCs significantly enriched in sensitized clones (Fig. 4e). Among these were TFs regulating stemness such as *Gata2*, *Meis1*, *Gfi1b* and *Sox4*. In contrast, several loci related to differentiation (*S100a8*, *F10*, *Timp2*) were enriched in de-sensitized clones, suggesting that differentiation state of the cells (SCF vs GM-CSF) plays a role in determining the response to *Tet2* KO.

Next, in order to better understand if solely the differentiation state of cells can predict the degree of functional response to *Tet2* KO, we defined where each clone lies in the differentiation trajectory. To do so, we correlated chromatin profiles of WT clones to the GMP signatures identified in Fig. 1 (stem-like, neutrophilic, monocytic, Fig. 4f). Differentiated GM-CSF clones showed as expected, high monocytic/neutrophilic and low stem-like signatures. Focusing within SCF clones, we did not find a good correlation between sensitized clones and the GMP stem-like signature, consistent with other clonal features contributing to sensitivity to *Tet2* mutation.

We thus sought to better characterize heterogeneity among SCF clones. We performed combined scRNAseq and ATACseq analysis using SHARE-seq[40] on a subset of SCF GMP clones characterized by different functional properties. Visualization of some key motifs and marker genes allowed us to identify regions characterized by different levels of activity of key differentiation regulators (Fig. 4g). We then observed that the distribution of cells across different WT clones is not overlapping, but rather single clones accumulate in specific regions of the UMAP coordinates (Fig. 4f, Supplementary Fig. 6c). This result further validates that functional heterogeneity observed upon differentiation is preceded by extensive molecular priming at the level of progenitor cells. By considering the SHARE-seq profiles of the *Tet2* mutated clones, we observed a marked perturbation in cell state relative to their WT counterparts (Fig. 4f, Supplementary Fig. 6c). Clones showing high Functional Perturbation Score, such as clones 12 and 26, were enriched in a separate region and showed a marked shift in UMAP coordinates after *Tet2* KO as compared to other analyzed clones. By analyzing ATAC-seq tracks at the proximal promoter region of several transcription factors positively correlated with high *Tet2* sensitivity, we confirmed higher accessibility in hyper-sensitized clones (*Sox4*, *Meis1*, *Runx3*; Supplementary Fig. 6d). These data collectively reinforce the notion that within the same differentiation state extensive epigenetic heterogeneity is observed, and we identified a permissive epigenetic state correlated with higher molecular perturbation upon *Tet2* KO.

We next tested whether the increased activity of chromatin regulators positively correlated with high *Tet2* sensitivity from Fig. 4f could be functionally implicated to foster the permissive clonal state. Among those, *Sox4* was of particular interest given its known functional association with AML when overexpressed with multiple

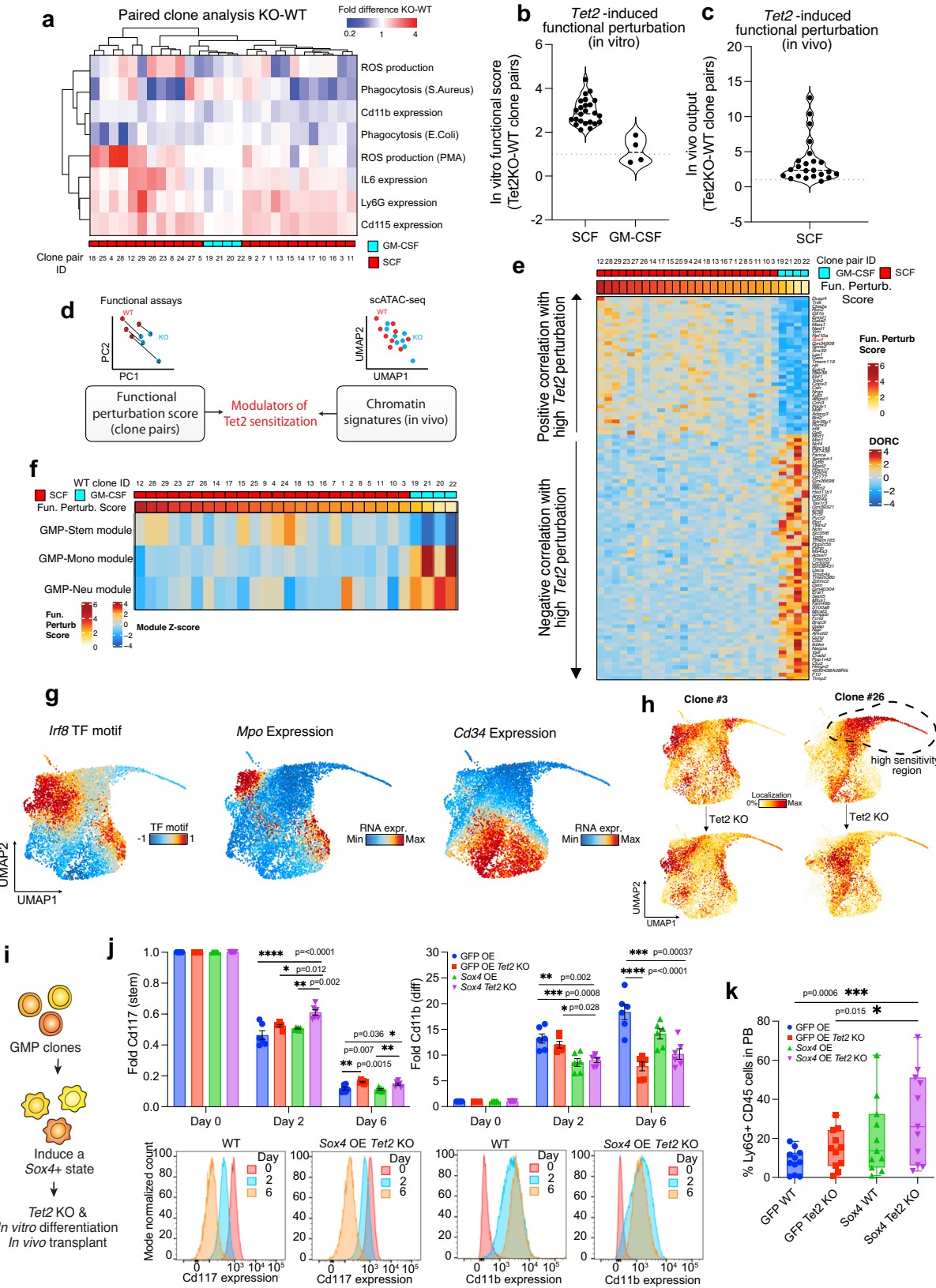

oncogenes[62–65]. Sox4 is a member of the SRY-related HMG-box TF family. Its expression is highest in primitive progenitors and progressively decreases with myeloid differentiation[63]. Its upregulation has been associated with increased stem cell activity and self-renewal[66,67] and blocking differentiation during leukemogenic transformation[68].

Here, to match physiologically-relevant levels of Sox4, we overexpressed its ORF using a lentiviral construct with mild promoter activity in hematopoietic cells[69](Fig. 4i). With this system, we achieved a 2-fold increase in Sox4 expression in GMP clones as compared to endogenous expression levels (Supplementary Fig. 6e). We measured the functional effects of Sox4 overexpression including a time course

**Fig. 4 | GMP clonal system defines sensitization status to *Tet2* mutation.**
**a** Paired analysis showing fold change difference between WT and *Tet2* KO clones tested in assays from Fig. 3c. Same as Fig. 3d but each single clone pair is shown. GM-CSF and SCF clones are color-coded. **b** Functional perturbation score measured for each WT and *Tet2* KO GMP sister clones, indicating the extent of functional alterations measured using in vitro assays. Dashed line:median. **c** In vivo output for each WT and *Tet2* KO GMP SCF clone pair, indicating the extent of functional alterations measured upon in vivo transplantation. Dashed line:median.
**d** Schematics depicting the analysis framework utilized to link clone phenotypes to in vivo chromatin states and identify epigenetic predictors of *Tet2* sensitization states. **e** Heatmap showing the top DORC chromatin regions correlated to WT GMP clones ranked by functional score. GM-CSF and SCF clones are color-coded.
**f** Heatmap showing correlation between WT GMP clones ranked by functional score and GMP chromatin signatures from Fig. 1. GM-CSF and SCF clones are color-coded.
**g** UMAP plots showing representative markers (RNA or ATAC) relevant for GMP

identity and differentiation (SHARE-seq data). **h** UMAP plots showing single cell distribution of SCF WT and mutated GMP clone pairs analyzed by SHARE seq.
**i** Schematics representing the experimental workflow for generating *Sox4*-overexpressing GMP clones and relative analyses. **j** Expression levels of Kit (left) and Cd11b (right) at different time points during GMP clone differentiation. Values are normalized to day 0. *n* = 3 clones in technical duplicates. One-way Anova with Turkey's multiple comparison test, Exact *p*-values are indicated on the figure. Data are presented as mean values +/- SEM. Corresponding representative flow plots are shown on the bottom panel. **k** In vivo output of GMP clones treated as indicated. PB analysis is performed at day 7 after GMP transplantation. *n* = 12 mice; data from 2 independent experiments are included. One-way Anova with Turkey's multiple comparison test, Exact *p*-values are indicated on the figure. Box: 25th to 75th percentile; line:median; whiskers: min to max. Source data are provided as a Source Data file.

of GMP differentiation using surface expression of c-Kit and CD11b. While *Tet2* KO resulted in a decrease of CD11b and an increase in c-Kit levels compared to control, concomitant overexpression of *Sox4* delayed GMP differentiation especially in the early time points (Fig. 4j). This differentiation delay is not due to reduced cell proliferation, but likely to increased self-renewal divisions, as *Sox4* overexpression increased proliferation of WT or *Tet2* KO GMP clones both in steady state and upon differentiation (Supplementary Fig. 6f, g). Finally, we measured GMP clonal output by mouse transplantation assays. Notably, the combination of Sox4 overexpression significantly increased the myeloid output of *Tet2* KO GMP clones in vivo (Fig. 4k).

Collectively, these data demonstrate that defined epigenetic states of clones can lead to divergent phenotypes in response to an identical genetic perturbation. Moreover, Sox4 activity promotes a sensitized cell state to *Tet2* which is characterized by stem-like properties.

## *Sox4* enhances epigenetic dysregulation observed after *Tet2* deletion

The GMP clonal system is used here as a discovery and screening tool to reliably study clonal properties and link epigenetic properties in progenitors to functional consequences of *Tet2* mutation. However, the target cell population of CH is more primitive. We thus independently validated and generalized our findings in primary HSC-based systems (Fig. 5a).

Primary KSL (Kit+ Sca-1+ Lin-) cells from *Tet2*[fl/fl] Mx-1 Cre mice were transduced with either GFP or *Sox4*-overexpressing lentiviral constructs and transplanted into lethally irradiated recipients in competition with unmanipulated CD45.1(STEM)[70] congenic cells. *Tet2* deletion was introduced 6 weeks after the transplant by pI:pC administration and the in vivo contribution was measured up to 24 weeks post-transplant. We observed a significant increase in the in vivo repopulation of *Sox4* OE *Tet2* KO cells as compared with *Tet2* KO alone (Fig. 5b). Mice transplanted with *Sox4* OE HSC did not show a repopulation advantage as compared to controls, validating that Sox4 specifically provides HSC competitive fitness only when in combination with *Tet2* deletion (Supplementary Fig. 7a). We then investigated if *Sox4* overexpression induced changes in the cell composition and showed that *Sox4* OE *Tet2* KO mice did not show any significant skewing in the distribution of the differentiated BM lineages (Supplementary Fig. 7b) and the Lin- progenitors (Supplementary Fig. 7c). Significant differences were instead observed in the distribution of KSL primitive cells, with an expansion of MPP4 cells (Fig. 5c). This population represents a highly proliferative direct precursor of both lymphoid and myeloid progenitors[71] and its accumulation is consistent with an exacerbation of impaired downstream differentiation observed in the *Tet2* KO context.

In order to characterize the molecular changes associated with the interaction between *Sox4* and *Tet2*, we further performed scATAC-

seq on BM populations with the same sorting strategy as in Fig. 1 (HSC to mature myeloid effector cells, *n* = 2 *Sox4* OE *Tet2* KO). Data were batch corrected and plotted on the same UMAP coordinates as Fig. 1 (Supplementary Fig. 1g-k). We then performed differential chromatin accessibility on the GMP subset and found 8282 peaks upregulated and 4704 peaks significantly downregulated in the *Sox4* OE *Tet2* KO condition relative to WT cells (Fig. 5d). Overlapping the differential peak signatures, *Sox4* OE *Tet2* KO included the vast majority of differential peaks observed in the *Tet2* KO alone samples (82% of downregulated and 91% of upregulated peaks) (Supplementary Fig. 7d). In addition, 6019 peaks were uniquely upregulated comparing *Sox4* OE *Tet2* KO condition relative to WT cells, consistent with an established role for Sox4 as a transcriptional activator[72]. Overlapping the differential peaks with the previously identified epigenetic signatures that define myeloid differentiation, demonstrated that *Sox4* overexpression drastically increased the epigenetic dysregulations observed upon *Tet2* KO (Fig. 5e), with upregulated peaks strongly enriched in the GMP-stem signature and downregulated peaks in GMP- mono/neutro signatures. This finding was also confirmed by plotting the accessibility of the GMP differential peaks on UMAP coordinates (Fig. 5f), showing that upregulated peaks have high accessibility in primitive cell types. These data strongly indicate that in the presence of high *Sox4* activity, *Tet2* KO GMPs are likely stalled at primitive stages of differentiation. Finally, this finding was also confirmed by the skewed distribution of *Sox4* OE *Tet2* KO GMP cells, which accumulate in the early progenitor stages and are prevented from entering into myeloid commitment (Fig. 5g, compare with Fig. 1f). Increased dysregulation of myeloid-related loci was also seen when calculating DORC accessibility scores across the differentiation trajectory (Supplementary Fig. 7e).

Overall, these data indicate that *Sox4* acts in a synergistic manner with *Tet2*, leading to increased phenotypic and epigenetic dysregulation.

## Cell reprogramming by *Sox4* induces a hypercompetitive HSC state in vivo

To understand the mechanism underlying the observed *Sox4*-mediated increase in stem cell properties and hyperresponsiveness to *Tet2*, we utilized the well-established, PVA-based HSC expansion method[73]. Thereby, we could gene modify and analyze HSC in vitro without extensively compromising their functional properties. Moreover, the inflammation associated with the inducible Mx1 model for *Tet2* deletion presented in Fig. 5 could be avoided.

Long term HSC (KSL+, CD150 + CD48- EPCR + ) from *Tet2*[fl/fl] mice, transduced with *Sox4* OE or control and then with Cre-expressing LV to induce Tet2 KO were cultured in PVA-based medium (Fig. 6a). As functional and molecular readouts, cells were analyzed by flow cytometry for HSC markers, harvested for SHARE-seq and transplanted in vivo. In accordance with our hypothesis, we observed that the combination of *Sox4* OE and *Tet2* KO resulted in a significant

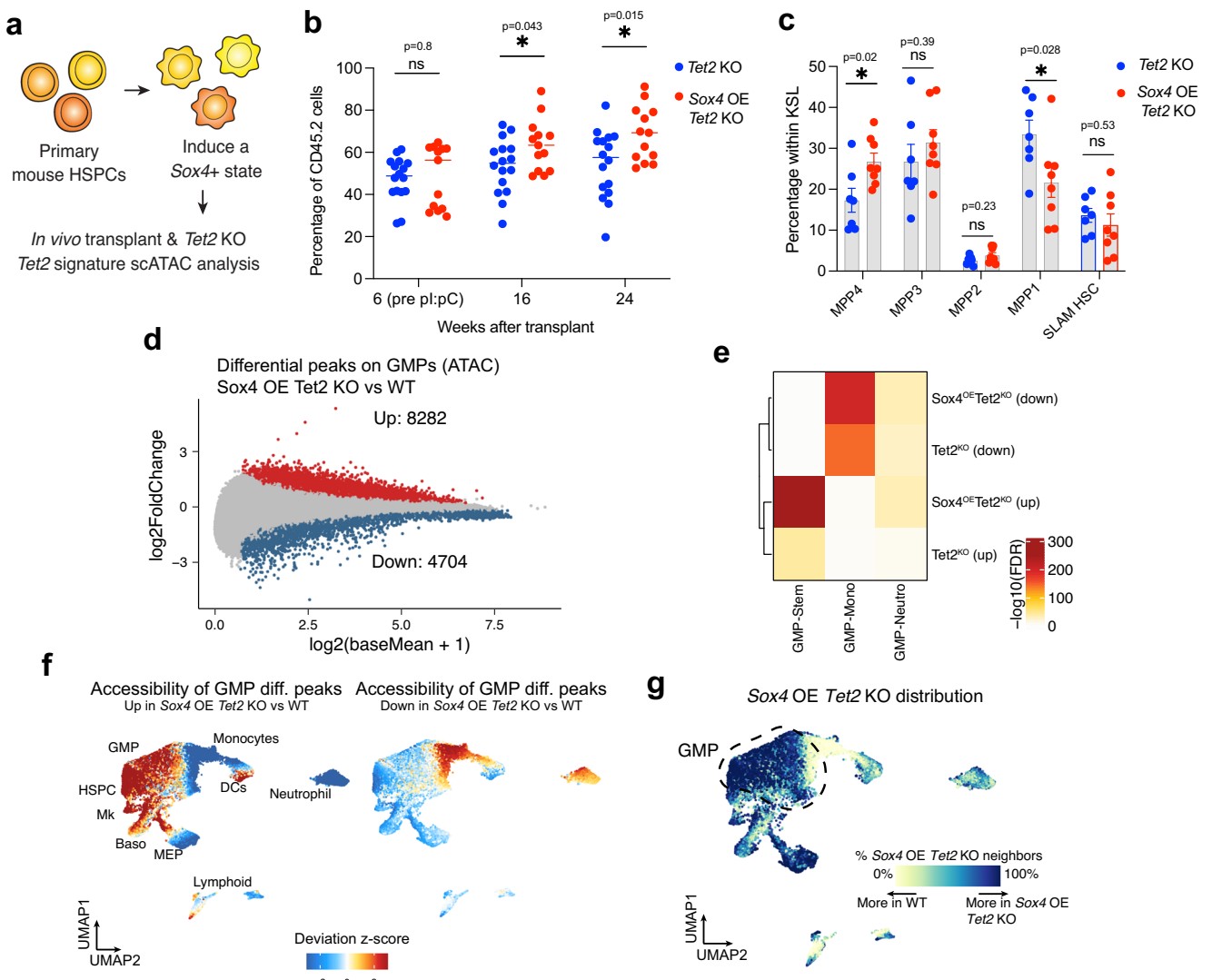

**Fig. 5 | *Sox4* enhances epigenetic dysregulation observed after *Tet2* KO in primary cells. a** Schematics representing the experimental workflow and analysis for generating *Sox4*-overexpressing primary HSPC in *Tet2*fl/fl background. **b** Percentage of CD45.2 cells measured at different time points in the PB of mice transplanted with inducible *Tet2* KO cells together with overexpression of GFP (*Tet2* KO, n = 16) or *Sox4* (*Sox4* OE *Tet2* KO, n = 13). Two-tailed unpaired *t*-test, Exact *p*-values are indicated on the figure. Data from 2 independent experiments are included. Line: Median. **c** Distribution of the indicated populations within Lin- cells in the BM of mice from b 3 weeks after transplant (*Tet2* KO, n = 7; *Sox4* OE *Tet2* KO, n = 8). Data are presented as mean values +/- SEM. Mann-Whitney test; Exact *p*-values are indicated on the figure. Populations are defined as in[71]. **d** Plot showing the

fold change in the accessibility of chromatin peaks comparing WT and *Sox4* OE *Tet2* KO GMP cells analyzed by scATAC-seq. Cells belonging to clusters 1,4,7,10,13 for GMP samples were included. Significantly (FDR < 0.01) upregulated peaks are colored in red while significantly downregulated peaks are colored in blue. **e** Plot showing the overlap between GMP chromatin signatures calculated in Fig. 1 and differential peaks in GMP comparing *Tet2* KO vs WT (same as Fig. 2e) and *Sox4* OE *Tet2* KO vs WT. Level of significance is color coded. **f** Cumulative accessibility of differential upregulated (left) and downregulated (right) GMP peaks from Fig. 5d plotted on UMAP coordinates. **g** Plot showing relative distribution of WT and *Sox4* OE *Tet2* KO cells across the differentiation trajectory. Source data are provided as a Source Data file.

accumulation of primitive progenitors (Fig. 6b). SHARE-seq analysis revealed molecular signatures highly shared with unmanipulated hematopoietic populations[74], which allowed us to annotate these primitive subsets (Fig. 6c, Supplementary Fig. 8a). Gene Set Enrichment Analysis within the HSC cluster demonstrated that among the most deregulated pathways in the presence of *Sox4* OE were Notch, FGF and PI3K (Fig. 6d, Supplementary Fig. 8b). These data suggest pleiotropic effects which might promote growth[75–77]. Accordingly, direct comparison of *Sox4* OE *Tet2* KO cells vs *Tet2* KO cells showed differential expression of *Fgfr2*, *Camk1d* and *Cdk8* genes in HSC and committed compartments (Supplementary Fig. 8C). Corresponding scATAC-seq analysis revealed mild changes in accessibility of TF motifs which again pointed to enhanced cell growth of primitive compartment, as marked by increased activity of Atf3[78] and reduced differentiation capacity at

the level of progenitors (as marked by increases in Gata factors and decreased Spib/c) (Supplementary Fig. 8d). To gain more understanding of the molecular bases for how the *Tet2*-hypersensitive cell state induced by *Sox4* can alter cell phenotype, we performed untargeted metabolomic analyses on KSL cells sorted from transplanted mice (Fig. 6e). Previous studies have shown that *Sox4* and *Cdk8* expression correlate with changes in cell metabolism, in particular promotion of a glycolytic state[79,80]. Notably, we observed greater glycolysis in *Sox4* OE *Tet2* KO hypersensitive cells. GMP clones were utilized to confirm these findings since unbiased mass spectrometry methods benefit from higher cell input. We observed that *Sox4* overexpression in combination with *Tet2* KO increased metabolites associated with purine metabolism, TCA cycle, Phosphoinositide (PI) signaling and glycolysis (Supplementary Fig. 8e). This is consistent

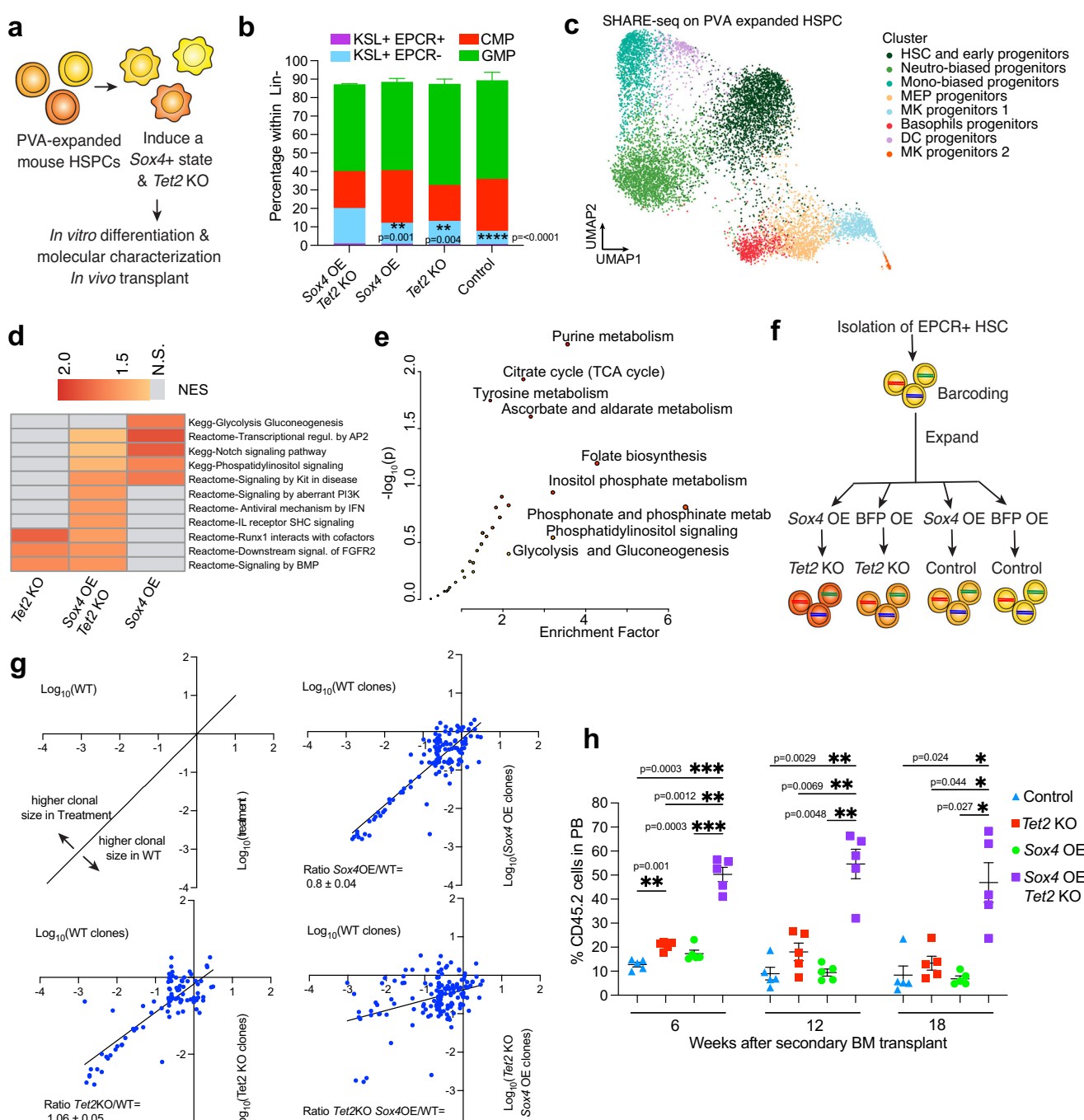

**Fig. 6 | *Sox4*- induced cell state provides competitive advantage to *Tet2* KO primitive cells. a** Schematics representing the experimental workflow for generating PVA-expanded HSPCs (*Sox4* OE and *Tet2* KO) and relative molecular analyses. **b** Flow cytometry analysis showing the distribution of the indicated primitive population after prolonged expansion of mouse HSC with PVA-containing medium for 3 weeks. *n* = 4, Data from 2 independent experiments are included. One-way Anova with Turkey's multiple comparison test; Exact *p*-values are indicated on the figure. Data are presented as mean values +/- SEM. **c** Unsupervised clustering annotation of in vitro expanded mouse HSPC and analyzed by SHARE-seq (**d**) Selected pathways from GSEA analysis comparing gene expression data from HSC cluster from Fig. 6c. NES: Normalized enrichment score (**e**) Untargeted metabolomic analysis showing deregulated pathways comparing *Sox4* OE *Tet2* KO vs WT KSL HSC harvested

from mice, *n* = 3. Statistical values obtained using Metaboanalyst 5.0 software[118] (**f**) Schematics representing the experimental workflow for paired clonal tracking study on PVA-expanded HSPCs. **g** Clone size comparison of PVA-expanded HSC sister samples treated as indicated. Clone pairs *Tet2* KO vs WT *n* = 97, *Sox4* OE vs WT *n* = 130, *Sox4* OE *Tet2* KO vs WT *n* = 158. Blue line represents linear regression and highlights relative expansion of *Sox4* OE *Tet2* KO clones as compared to WT counterparts, (**h**) Percentage of CD45.2 cells measured at different time points in the PB after secondary BM transfer of cells from 6b. *n* = 5, Data from 1 independent experiment is included. Two-way Anova with Turkey's multiple comparison test, Exact *p*-values are indicated on the figure. Data are presented as mean values +/- SEM. Source data are provided as a Source Data file.

with the scRNA-seq analysis performed on PVA-expanded HSC, which also indicated an increase in glycolytic gene expression (Fig. 6d). Intriguingly, among the most upregulated metabolites with *Sox4* overexpression was Hydroxyglutaric acid (2HG) (Supplementary Fig. 8f), an oncometabolite that directly inhibits Tet dioxygenases[81]. Absence of *Tet1/Tet3* activity in combination with *Tet2* KO was indeed reported to accelerate the leukemogenesis process[82] and their indirect inhibition by *Sox4* may contribute to the highly increased epigenetic dysregulation observed in *Sox4* OE *Tet2* KO cells.

To test whether Sox4 fosters a hyperproliferative state at a clonal level in primary HSC, we barcoded the cell population using a high-diversity clonal tracing library (Fig. 6f). After transduction and 7 day PVA expansion the population was split into sister aliquots before introduction of *Sox4* OE and *Tet2* KO thereby enabling paired clonal analysis[83]. Comparing the size of paired sister clones across the different treatments, we observed that only the combination of *Sox4* OE and *Tet2* KO is able to provide a hyperproliferative advantage compared to WT. Interestingly, only a subset of clones showed this effect further confirming clone-specific behaviors in response to the same genetic modification (Fig. 6g).

Finally, to validate our findings in vivo, we performed competitive transplantation into lethally irradiated mice and compared the hematopoietic output of the genetically modified PVA-expanded HSC. A significant increase in repopulation ability of *Sox4* OE *Tet2* KO cells was observed on secondary transplantation (Fig. 6h). While no changes were noted on primary transplant (Supplementary Fig. 8g), secondary transplant is a reliable assay for HSC self-renewal.

These data demonstrate that *Sox4* activity alters cell sensitivity to *Tet2* mutation in a pleiotropic manner affecting proliferation, differentiation and metabolism to collectively create a cell state of increased competitive advantage.

## Discussion

Somatic mutations in epigenetic regulators have been extensively characterized in mouse models and human samples as drivers of CH. Two questions that are less well defined are approached here.

The first is a fuller definition of the downstream molecular consequences of the mutations which enable clonal outgrowth. It is well known that mutant *TET2* alleles are associated with significant expansion of the myeloid compartments[30]. Here, we defined the multi-omic landscape that underlies differentiation of primitive progenitors towards myeloid lineages and how this network is disrupted in a relevant *Tet2* KO mouse model. In accordance with previous datasets[32,84], we described that mutant cells display highly overlapping transcriptional and epigenomic profiles as compared to WT controls, highlighting that Tet2-mediated regulation doesn't globally affect cell identity but rather induces functionally relevant cell states with altered differentiation properties. Consistently, our data indicate that Tet2 regulation is directly required for priming of GMPs towards mature myeloid lineages and in its absence cells accumulate markers of early primitive progenitors such as *Runx1*, *Gata2*. This finding might provide a molecular mechanism behind the expansion of GMPs which is observed after *Tet2* KO and provides an enlarged reservoir of progenitors which feeds the myeloproliferation observed, especially during inflammatory conditions[47]. The functional alterations we found at the level of effector granulocytes or monocytes foster inflammation and decrease pathogen clearance potentially enforcing a positive feedback loop.

The second question is how single clones are differentially impacted by a genetically identical mutation. Our data indicate that selected epigenetic states can contribute to clone sensitivity to *Tet2* mutations. By using a clonal GMP system, we were able to correlate the extent of functional changes measured in KO clones to corresponding molecular properties measured in WT counterparts. We demonstrated that chromatin states with features of stem-like cells can prime

myeloid progenitors to stronger functional alterations. These data correspond with our finding that *Tet2* regulates stem cell and early progenitor identity before lineage priming, the same setting thought to be the cell of origin in the transition from CH to MDS/AML[85]. We identified *Sox4* as a driver of hyper-sensitization state towards *Tet2* inactivation. Increased *Sox4* activity is reported to synergize with oncogenes during the process of leukemic transformation[62-65]; its expression peaks in primitive cells and gradually decreases along myeloid differentiation[36]. Moreover, Sox4 expression is increased when its direct negative regulator *CEBPA* is mutated. Clinical data show that *TET2* mutations are frequent in *CEBPA* double allelic mutant AML patients, and those cases are associated with worse prognosis[86]. Our data are consistent with a model where *Sox4* promotes the maintenance of an immature and hyperproliferative cell state, which increases repopulation advantage and likely the susceptibility of accumulation of secondary mutations. Importantly, this concept was also validated using a clonal tracing method in primary HSC.

We provide evidence that *Sox4* acts through two mechanisms, (i) by directly activating transformation-promoting pathways such as Notch, FGF and PI3K and (ii) by potentiating Tet2-mediated epigenetic dysregulation. Increased expansion of *Sox4*, *Hlf* and *Meis1* expressing cells has also been detected in another independent study utilizing inducible *Tet2* KO mice[32], reinforcing our findings.

Our HSC models highlight also that high *Sox4* expression is associated with alterations in genes involved in cell metabolism such as *Cdk8*, and untargeted metabolomics findings reinforced this finding. We hypothesize that induction of altered glycolytic states by *Sox4* might play a role in enhancing proliferation/self-renewal of primitive progenitors in CH as described in non-malignant contexts[87] and during AML development[88]. We thus speculate that metabolic reprogramming may not only be a hallmark of cell transformation but also play an active role in the selection/predisposition of pre-malignant clones for disease development.

Furthermore, our data highlight that individual cell types can be affected by *Tet2* differently. Although widely expressed, *Tet2* interaction with DNA relies on the interaction with chromatin modifiers and TFs, due to the lack of a direct DNA binding domain. This interaction enables a cell type and chromatin context-specific activity[89]. In hematopoietic cells, CH studies described preferential recruitment of Tet2 at distal enhancer regions in myeloid cells, where it co-localizes with TFs involved in lineage specification and differentiation (Erg/Etv, Runx1-3, Gata1-3, Cebpa/b)[45]. The pleiotropic effects of *Tet2* KO among different cell types could be thus explained by differential availability and DNA binding capacity of its interactors. Methylation in the binding site (and thus reduced DNA accessibility) of key Tet2 interactors has been proposed as a mechanism to explain incorrect lineage differentiation in *Tet2* deficient human cells[90]. Similarly, *Tet2* mutation was reported to synergize with inactivating mutations in key interactors such as *PU.1* for leukemic transformation[91], highlighting the concept that availability of Tet2 interactors might also play a role in determining the downstream functional impact of the mutation.

Our study has implications regarding the definition of risk criteria for CH patients. Recent studies exploiting combined single-cell allele genotyping with transcriptomics/epigenomics analyses[84,92,93] have characterized the landscape of human CH mutant cells. Also, the identification of clonal identities within mutant cells has been accomplished with additional integration of clone-specific molecular identifiers, such as mitochondrial or somatic DNA mutations[94-97]. While these studies highlighted some key molecular differences of mutant populations or clones at different stages of hematopoietic transformation, they have not been able to identify molecular factors that may facilitate the transition from CH to MDS/AML. Here, by using a prospective approach, we uncovered Sox4 expression levels as a specific molecular target implicated in increasing clonal selective advantage

and expansion after introducing a CH mutation. Recent studies indicate that the mechanism could be conserved in humans and relevant for disease progression. In *TET2* mutated AML samples, aberrant CTCF sites are enriched in *SOX4* motifs specifically and pathway analysis showed enrichment for NOTCH and WNT signaling[98]. To formally validate this finding in human cells, longitudinal analysis of this biomarker in single CH clones will be required.

The contribution of the epigenome to overt cancer evolution has recently emerged as a highly significant concept[99,100]. A recent study demonstrated that clonally stable and heritable chromatin alterations drive colorectal cancer, and progression is associated with reactivation of homeobox genes[101] and multiple studies highlighted its relevance in progression to blood malignancy after the introduction of different genetic mutations[102–104]. Our work extends this concept to premalignant and pre-mutation states and specifically defines that the epigenetic features of the clone before a mutation occurs can determine the phenotype acquired. For CH, this finding can provide insight into the incomplete penetrance for disease development observed in patients. Although the specific CH driver mutation will likely determine the specific mechanisms of transformation, heterogeneity at the level of the clone of origin (within HSC and early progenitor compartments) could affect long-term expansion or facilitate further clonal selection and acquisition of secondary mutations. Further, we identified *Sox4* activity as facilitating *Tet2*-related molecular alterations that foster clonal outgrowth in vivo. Our results validate the hypothesis that epigenetic features can predispose mutant hematopoietic clones for transformation and underscore the importance of pursuing prospective clonal tracking studies for the identification of such risk states for CH malignant evolution.

## Limitations of the study
A potential limitation of our study is the use of an in vitro clonal system for the identification of *Tet2*-related functions. This model may be suboptimal mainly because (i) cells are conditionally immortalized using HoxB8, which could influence chromatin accessibility and alter regions bound by Tet2 and (ii) cells are arrested at the GMP state, therefore missing information regarding more primitive progenitors relevant for disease progression. Complementary in vivo studies using clonally tracked HSC-based models such as recently described[27] could allow uncovering of other molecular players that determine clonal advantage in CH models. Moreover, a clear distinction between phenotypic effects driven by differentiation stages or heterogeneous epigenetic states within the same cell type can be difficult to determine as these two cellular properties are closely related.

## Methods
### Mice
*Tet2*[fl/fl] mice[30] (#017573, obtained from The Jackson Laboratory) were crossed to Mx1-Cre mice (#003556, obtained from The Jackson Laboratory). Mx1-Cre negative littermates were utilized as controls. Genotyping was performed with primers listed in Supplementary Data 4 (WT amplicon: 250 bp, fl/fl amplicon: ~450 bp). For competitive bone marrow transplantation experiments, aged-matched CD45.1(STEM) mice[70] were utilized (mice were bred at Massachusetts General Hospital). WT C57BL6/J recipients (#000664, obtained from The Jackson Laboratory) were subjected to whole body irradiation (9.5 Gy) from a 137Cs source 1 day before BM transplantation. 4 doses of 12.5 μg/g pI:pC (Amersham) were administered intraperitoneally to induce Cre activity. Eight to twelve weeks old, age-matched randomized male and female mice were used in all experiments. For bone marrow transplantation experiments, female mice were used as recipients. All mice were bred and maintained in pathogen-free conditions and all procedures performed were approved by the Institutional Animal Care and Use Committee of Massachusetts General Hospital (protocol #2016N000085).

### Cell lines
Hoxb8-ER conditionally immortalized clones were generated at Massachusetts General Hospital and maintained in RPMI medium (Thermo Fisher) supplemented with 1% Penicillin/Streptomycin, 1% Glutamine, 0.5 μM beta-estradiol (Sigma, E2758) and conditioned media containing different cytokines. For the neutrophil-biased cells, media contained ~100 ng/ml SCF (generated from a Chinese hamster ovary cell line that stably secretes SCF), whereas for macrophage progenitors GM-CSF 10 ng/ml (Peprotech). Hoxb8-ER cultures were kept in a 5% $CO_2$ humidified atmosphere at 37 °C.

293 T cell line used for retroviral and lentiviral production was obtained from ATCC (CRL-3216).

### Primary cells
For BM transplant studies, primary mouse Lin- Kit+ Sca+ cells were seeded at the concentration of $5 \times 10^5$ cells/ml in serum-free StemSpan medium (StemCell Technologies) supplemented with 1% Penicillin/ Streptomycin, 1% Glutamine, and mouse early-acting cytokines (mSCF 100 ng/ml, m Flt3-L 100 ng/ml, mTPO 50 ng/ml, and mIL-6 20 ng/ml; all purchased from Peprotech). For PVA-expansion studies, Lin- Kit+ Sca+ Cd150 + Cd48- Epcr+ cells were seeded in 1X Ham's F-12 Nutrient Mix liquid media (Gibco), supplemented with 1 M HEPES (Gibco), 1% Penicillin/Streptomycin/Glutamine, 1% ITSX (Gibco), mSCF 10 ng/ml, mTPO 100 ng/ml (Peprotech), 1 mg/ml PVA (Millipore Sigma). Further details are provided in the following sections. HSC cultures were kept in a 5% $CO_2$ humidified atmosphere at 37 °C.

### scATAC-seq single cell profiling
Cell lysis, tagmentation and droplet library preparation were performed following the SureCell ATAC-Seq Library Prep Kit User Guide (17004620, Bio-Rad). Harvested cells and tagmentation related buffers were chilled on ice. Lysis was performed simultaneously with tagmentation. Washed and pelleted cells were resuspended in Whole Cell Tagmentation Mix containing 0.1% Tween 20, 0.01% Digitonin, 1 x PBS supplemented with 0.1% BSA, ATAC Tagmentation Buffer and ATAC Tagmentation Enzyme. Cells were mixed and agitated at 500 rpm on a ThermoMixer (Eppendorf) for 30 min at 37 °C. Tagmented cells were kept on ice prior to encapsulation. Tagmented cells were loaded onto a ddSEQ Single-Cell Isolator (12004336, Bio-Rad). Single-cell ATAC-seq libraries were prepared using the SureCell ATAC-Seq Library Prep Kit (17004620, Bio-Rad) and SureCell ddSEQ Index Kit (12009360, Bio-Rad). Bead barcoding and sample indexing were performed following the standard protocol and the number of amplification cycles was adjusted according to cell input. Libraries were loaded on a NextSeq 550 (Illumina) and sequencing was performed using the NextSeq High Output Kit (150 cycles) and the following read protocol: Read 1 118 cycles, i7 index read 8 cycles, and Read 2 40 cycles. A custom sequencing primer is required for Read 1 (16005986, Bio-Rad).

### scRNA-seq single cell profiling
scRNA-Seq was performed on a Chromium Single-Cell Controller (10X Genomics) using the Chromium Single Cell Reagent Kit v2, Chromium Next GEM Chip A and Chromium i7 Multiplex Kit according to the manufacturer's instructions. Briefly, single cells were partitioned in Gel Beads in Emulsion (GEMs) and lysed, followed by RNA barcoding, reverse transcription and PCR amplification (according to the available cDNA quantity). scRNA-Seq libraries were prepared according to the manufacturer's instructions, checked and quantified on Tapestation 4200 (Agilent) and Qubit 4 fluorometer (Invitrogen). Sequencing was performed on a Novaseq machine (Illumina) using the Novaseq S1 Kit (100 cycles).

### Multiplexed Bulk ATAC-seq
Indexed Tn5 transposome complexes were assembled as described previously[39] Also see this reference for a description of how the

barcodes were designed and a table with the oligo sequences. Cells were washed twice with 1 x PBS, counted, and resuspended to a concentration of $0.5 \times 10^6$ cells/mL. 2 µL of cells in 1 x PBS (1,000 cells) were mixed with 2 µL of barcoded Tn5, 5 µL 2x Illumina Tagment DNA Buffer (TD), 0.1 µL 10% NP40 (final concentration 0.1%) and 0.9 µL $H_2O$ in a 96 well plate. Each well contained a different sample and Tn5 barcode. Cells were mixed and agitated on a ThermoMixer (Eppendorf) at 500 rpm for 30 min at 37 °C. All the wells were pooled together on ice to prevent cross-contamination between Tn5 barcodes. Tagmented DNA was purified using a MinElute PCR Purification Kit (Qiagen), then minimally amplified for sequencing as previously described[105]. Final libraries were purified using the MinElute PCR Purification Kit (Qiagen), and sequenced on a NextSeq 550 (150 cycles), using the following parameters: Read 1 92 cycles, i7 index read 8 cycles, and Read 2 66 cycles, 50% of PhiX Sequencing Control. A custom sequencing primer is required for Read 1 (16005986, Bio-Rad).

## SHARE-seq

SHARE-seq was performed as described previously[40]. Briefly, single cells were fixed by adding Formaldehyde (28906, ThermoFisher) at a final concentration of 1%. Fixed cells were transposed using barcoded Tn5 (Seqwell) in a transposition buffer (1 x TD buffer from Illumina Nextera kit, 0.1% Tween 20 (P9416, Sigma), 0.01% Digitonin (G9441, Promega)) at 37 °C for 30 minutes with shaking at 500 rpm. Transposed cells were reverse transcribed using Maxima H Minus Reverse Transcriptase along with RT primer containing a Unique Molecular Identifier (UMI), a universal ligation overhang and a biotin molecule. Ligation of barcoded adapters was performed using three rounds of split pool barcoding followed by reverse crosslinking. ATAC and RNA libraries were prepared as previously described[40]. Libraries were quantified with KAPA Library Quantification Kit and pooled for sequencing. Libraries were sequenced on the Nova-seq platform (Illumina) using a 200-cycle S1 kit and the following read protocol: Read 1: 50 cycles, Index 1: 99 cycles, Index 2: 8 cycles, Read 2: 50 cycles.

## scATACseq data processing

Genome-wide chromatin accessibility peaks were called using MACS v2 (MACS2)[106] on the merged aligned scATAC-seq reads per condition, generating a list of peak summit calls per condition. To generate a non-overlapping set of peaks, we first extended summits of each condition to 800 bp windows (±400 bp). We combined these 800 bp peaks, ranked them by their summit significance value and retained specific non-overlapping peaks on the basis of this ordering. We further added to the peak list all non-overlapping peaks from the ImmGen ATAC-seq atlas, after also extending the ImmGen peaks to 800 bp windows[107] (https://sharehost.hms.harvard.edu/immgen/ImmGenATAC18_AllOCRsInfo.csv). This resulted in a filtered list of disjoint peaks ($n = 297,361$), which were finally resized to 301 bp (i.e. ±150 bp from each peak summit) and used for all downstream analyses.

## scRNA-seq analysis

Base call files were demultiplexed, for each flow cell directory, into FASTQ files using Cellranger v3.1.0 (https://github.com/10XGenomics/cellranger) mkfastq with default parameters. FASTQ files were then processed using Cellranger count with default parameters. Gene-mapped counts were then loaded into R as a Seurat[108] object and used for downstream analysis. Genes with at least one UMI across cells were retained, and cells with a number of unique feature counts ≥ 200 and total UMIs ≥ 5000 and mitochondrial read percentage of <5% were initially retained. Normalization and scaling of RNA gene expression levels was then performed. PCA dimensionality reduction was run and UMAP was used for the final 2D cell projection (top 30 PCs). A cell kNN graph was determined using the FindNeighbors function in Seurat ($k = 30$ cell neighbors). Cells were then grouped into clusters using the FindClusters Seurat function (resolution = 0.8; Leiden algorithm), and

cluster and cell annotations manually assigned by visualizing the mean and percent expression of cell identity markers within cell clusters. Broader annotations were determined by merging finer cell groupings.

## scATAC-seq single cell clustering and annotation

First, dimensionality reduction was performed with cisTopic[109] using the runWarpLDAModels function as part of the cisTopic R package, with the prior number of topics set to 50. Next, Harmony[110] was run on the cisTopic cell Z-scores to adjust for observed sequencing batch effects (correcting for animal as a batch covariate). The batch-corrected cisTopic cell Z-scores were then used to project cells in 2D by running UMAP as part of the uwot R package, with $k = 50$ cell neighbors and a cosine distance metric. Cells were clustered using a Louvain algorithm, and clusters were annotated using gene activity scores and gene expression markers (see below).

## Cell localization analysis

For visualization of sample distribution on UMAP coordinates (Figs. 1f, 5g), the % cell neighborhood per cell that belongs to a specific sample was represented for *Tet2* KO and *Sox4* OE *Tet2* KO cells (relative to wild type cells). For each cell, the $k$-nearest neighbors were considered ($k = 50$) using the batch-corrected (harmony) principal components, and the fraction of neighborhood cells that came from a non-wild-type genotype was determined, and shown on the UMAP. All mice were utilized for this analysis.

Similar analysis was performed for Fig. 4g and Supplementary Fig. 6c, representing the % cell neighborhood per cell that belongs to a specific sample for each SCF-derived GMP clone (relative to all other clones).

## TF motif scores

TF motif accessibility Z-scores were computed for scATAC-seq data using chromVAR[111]. Briefly, scATAC-seq data ($n = 57,232$ cells; $n = 297,361$ peaks) was used as input, and GC bias for each peak was determined using the BSgenome.Mmusculus.UCSC.mm10 reference genome. Mouse cisBP TF motifs ($n = 890$ TFs) were then matched against the reference peak set, and $n = 100$ background iterations were used, using which deviation Z-scores were estimated using chromVAR's computeDeviations function.

## Gene TSS activity scores

Gene activity scores based on chromatin accessibility were derived for scATAC-seq data ($n = 57,232$ cells) using a sum of accessibility fragment counts around gene transcription start sites (TSSs), weighted inversely to the distance from the TSS, as previously described[42]. Aligned scATAC-seq fragments per cell are weighted based on the inverse distance to gene TSSs, then summed across the chosen window (9,212 bp) reflecting 1% of the total weight for the chosen exponential half-life (1 kb). Gene activity scores were then normalized by dividing by the mean score per cell, and used for downstream analysis.

## Differential peak testing

Differential testing of accessibility peaks was determined using DESeq2[112], as previously explored as a robust tool for analyzing ATAC datasets[113]. First, only early progenitor cells (excluding terminally differentiated clusters) were retained for differential accessibility signature derivation in GMP cells. This includes cells belonging to clusters 1,4 (only considering GMP sample), 7,10,13 ($n = 22,569$ cells). Next, for each annotated cell type, cell peak counts were "pseudobulked" or grouped per mouse sample, annotated cell type and genotype (WT or *Tet2* KO) by summing raw accessibility counts per peak per grouping. DESeq's negative binomial Wald test was then applied with default parameters, adjusting for celltype as a covariate, yielding estimates of fold-change and significance per peak between *Tet2* KO and WT samples. Only peaks with adjusted $p$-value ≤ 0.01 were retained as

differentially accessible between the two genotype groups, and were used as signatures for downstream analysis (e.g. TF motif enrichment testing and overlap with Tet2 ChIP-seq peaks). Same analysis was repeated for the *Sox4* OE *Tet2* KO group. These peak signatures were also used as peak annotation features with chromVAR to score single cells for KO signature accessibility relative to background peaks (Fig. 2b).

## Differential DORC analysis

For differential testing of DORC accessibility scores, we used normalized single cell DORC scores and performed differential testing using a Wilcoxon rank-sum test per cell type, comparing each *Tet2* KO condition to its control condition. FDR was determined to adjust for multiple tests. Cells belonging to clusters 1, 4, 7 (derived from GMP sorted sample) were utilized for this analysis.

## Co-accessibility modules

Chromatin accessibility peak "modules" were derived as previously described[42]. Briefly, TF motif accessibility Z-scores for cells that were annotated as wild-type (WT) GMPs ($n = 8,313$ cells) were used. First, TF motifs ($n = 890$; see TF motif scores section above) were clustered using a sequence similarity correlation cut-off $= 0.8$, and then the most variable motif in each cluster was used as a representative TF. Additionally, jackstraw PCA was performed on motif deviation Z-scores to determine TF motifs with significantly variable accessibility using $n = 100$ iterations and a jackstraw permutation $P \leq 0.05$ across the first 10 PCs. This yielded $n = 68$ significantly variable TF motif groups. Then, difference in mean accessibility was tested across for all reference peaks ($n = 297,361$) using normalized scATAC-seq peak counts (mean-centered per cell) between the TF high vs low cell groups (cell high/low groups divided based on the median Z-score across cells for each TF motif), and significant peaks for any given TF (FDR $\leq$ 1e-06 two-tailed $t$-test) were retained. Finally, fold-changes of mean accessibility between the high and low groups were used to cluster peaks into co-accessible modules, using a Louvain algorithm for determining communities ($k = 30$ peak nearest-neighbors), resulting in distinctly grouped peak modules, which were manually annotated based on the TF motifs that positively or negatively associated with their activity. These were then converted into a reference peak x module binary annotation matrix, and chromVAR was used to compute module accessibility deviation Z scores for each cell in the entire scATAC-seq dataset.

## scATAC-scRNA-seq cell pairing and visualization

Cells were paired between the two modalities using the scOptmatch workflow described previously[114]. Briefly, CCA was first run using Seurat's RunCCA function[108] to co-embed scATAC-seq and scRNA-seq data (using the cell KNN-smoothed normalized scATAC-seq gene TSS activity scores and scRNA-seq gene expression, respectively). Only the union of the top 5,000 variable gene scores (ATAC) and gene expression (RNA) was used to derive the top 30 CCA components, with rescaling of features performed prior to running CCA. These components were then used to pair cells between the two assays based on the minimum geodesic neighbors between ATAC-RNA cells across the entire data. For scATAC-seq cells, gene expression of paired scRNA-seq cells were then used to visualize gene expression markers on the scATAC-seq UMAP.

## Peak-gene cis regulatory correlation analysis

Peak-gene links and domains of regulatory chromatin were determined using the FigR R package[114]. Briefly, using paired scATAC and scRNA-seq data, we determined for each gene a set of *cis*-regulatory peaks that are most correlated with the given gene's expression. To do this, we tested peaks falling within 10 kb of a given gene TSS for correlation in peak accessibility and paired gene expression across single

cells, using $n = 100$ permuted background peaks (matched for peak GC-content and mean accessibility) for significance testing. Only peak-gene links with a positive correlation and permutation $P \leq 0.05$ were retained. DORCs were defined as genes having $\geq 3$ significantly associated peaks.

## Pathway enrichment analysis/GSEA

Enrichment analysis was performed using Metascape[115] using the following ontology sources: GO Biological Processes, GO Molecular Functions and Reactome Gene Sets. All genes in the genome have been used as the enrichment background. Terms with a *p*-value $< 0.01$, a minimum count of 3, and an enrichment factor $> 1.5$ are reported. The most statistically significant term within a cluster is chosen to represent the cluster. For analysis in Supplementary Fig. 1g, gene scores with correlation $> 0.1$ to each gene module were considered. For analysis in Fig. 2f, differential genes with $p$ value $< 0.001$ were considered. For GSEA[116], we performed pre-ranked analysis (genes ordered by fold-change) using the default settings. MSigDB Hallmark(H), KEGG and Reactome (C2) databases were utilized, pathways FDR $< 0.25$ were considered significant.

## Vectors, plasmids and molecular analyses

Lentiviral constructs expressing GFP, codon-optimized *Sox4*, or Cre recombinase were cloned into self-inactivating transfer constructs under the expression of human PGK promoter. Lentiviral backbones were obtained through MTA with Naldini lab (SR-TIGET, Milan). Lentiviral vectors were generated using HIV-derived, third-generation plasmids. Stocks were prepared and concentrated as previously described[117] using HEK-293T(ATCC, CRL-3216) as packaging cells. Titration was performed by qPCR[118] using a HEK-293T line with known number of vector integrations per diploid genome as standard. High complexity barcoding library LARRY Barcode Version 1 library[83] was a gift from Fernando Camargo (Addgene #140024). Retroviral stocks were generated using pCL-Eco plasmid, which was a gift from Inder Verma (Addgene plasmid # 12371).

For molecular analyses, genomic DNA was isolated with QIAamp DNA Micro Kit (QIAGEN) according to the number of cells available. Successful deletion of *Tet2* by Cre recombinase was measured from the gDNA by ddPCR (QX200, Biorad), quantifying the number of *Tet2* alleles relative to an unrelated genomic locus (*Sema3a*).

For gene expression analyses, total RNA was extracted using RNeasy Plus Micro Kit (QIAGEN), according to the manufacturer's instructions and DNase treatment was performed using RNase-free DNase Set (QIAGEN). cDNA was synthesized with SuperScript VILO IV cDNA Synthesis Kit (Invitrogen) and analyzed on a CFX Connect Real-Time PCR System (Biorad). The relative expression of each target gene was first normalized to housekeeping genes and then represented as 2^-DCt for each sample.

Sequences of DNA oligonucleotides used in this study are reported in Supplementary Data 4.

## Generation of Hoxb8-ER clones

Immortalization of murine BM cells with Hoxb8-ER was done as previously described[53] with the following modifications. Filtered BM cells were layered over Ficoll-Plaque-Plus (GE Healthcare Biosciences), and centrifuged at 400 x g for 25 min at RT without break to enrich for mononuclear cells. Cells were incubated in a 6-well tissue culture plate for 48 h at 37 °C with 5% $CO_2$ prior to the retroviral transduction in RPMI (RPMI-1640, Corning) with 10% 2 mM L-gluta-mine, and 100U penicillin/streptomycin (all from Thermo Fisher), supplemented with 20 ng/ml stem cell factor (mSCF), 10 ng/ml interleukin-3 (mIL-3), and 10 ng/ml interleukin-6 (mIL-6).Non-adherent cells were harvested and $5 \times 10^5$ cells were plated onto a 12-well tissue culture plate (Corning) coated with 10 mg/ml human fibronectin (Sigma). 1 ml of ecotropic retrovirus encoding MSCVneo-HA-

ER-Hoxb8 was applied in the presence of 8 mg/ml polybrene and spinoculation was performed at 1000 x g for 60 min at RT. After transduction cells were maintained in RPMI supplemented with 0.5 μM beta-estradiol (Sigma, E2758) and conditioned media containing ~100 ng/ml SCF for the generation of neutrophil-biased cells or GM-CSF 10 ng/ml (Peprotech) for the generation of macrophage progenitors. Antibiotic selection was performed by adding G418 at 1 mg/ml final concentration until untransduced control cells the control were not viable (usually ~ 7 days). After selection, cells were FACS sorted into 96 well plates with culture media for the generation of single cell clones.

For generation of *Tet2* KO lines, each clone was transduced with integrase-defective lentiviral vector encoding for Cre recombinase or GFP as control at a Multiplicity of Infection (MOI) of 200.

## Functional assays on myeloid cells

Cytospin and Wright-Giemsa staining: Cells were prepared in PBS at a concentration of $1 \times 10^6$ cells/ml and spun 1,000 RPM for 60 s on microscope slides. After air drying for 30 min, slides were sequentially soaked in different dilutions Wright-Giemsa stain (Siemens, 100% 4 min, 20% 12 min, 3X rinse in ddH2O). Coverslips were affixed with Permount Mounting Media (Thermo Fisher) and samples were analyzed at 100X magnification using oil immersion objective.

Phagocytosis assay: Cells were pre-stimulated with 100 ng/ml LPS (L2630, Sigma) for 30 min, washed in PBS and incubated in Live Imaging Solution (Thermo Fisher) along with labeled E. coli or S. aureus BioParticles (Thermo Fisher, 500 μg/ml and 1000 μg/ml, respectively) for 1 h at 37 °C before flow cytometry analysis.

Reactive Oxygen Species (ROS) assay: Cells were incubated using Invitrogen™ CellROX™ Flow Cytometry Assay Kit (Thermo Fisher) at 37 °C for 30 min in culture media, with or without 80 nM PMA (MIllipore Sigma).

Intracellular staining for cytokines: Cells were pre-stimulated with 100 ng/ml LPS together with protein transport inhibitor Golgi Plug (1:1000, BD) for 1 h at 37 °C. Surface and intracellular staining was performed using Perm/Fix kit (BD) according to manufacturer's instructions.

Proliferation assay: Cells were stained with 0.5 μM CellTrace Far Red Cell Proliferation Kit (Thermo Fisher) according to manufacturer's instructions. Flow cytometry analysis was performed 3 days later.

In vivo transplant: $2 \times 10^6$ Hoxb8-ER GMPs were transplanted in lethally irradiated recipients (9.5 Gy) together with $2 \times 10^5$ supporter CD45-mismatched BM cells via retro-orbital injection. Peripheral blood was collected at 4-7-9-11 days after injection.

## Functional perturbation score

Raw data outputs from obtained from functional assays performed in WT and *Tet2* KO GMP clones (expression of Ly6G, expression of CD115, expression of CD11b, expression of IL6, ROS production, phagocytosis after exposure to E.Coli and S.Aureus particles) were first standardized in order to render them comparable using the R function *scale*. A distance matrix among the different clones was then calculated using the R *dist* function and the *euclidean* distance measure. Functional perturbation score is then defined for each GMP clone as the distance between WT-KO paired clone pairs.

## Bone marrow transplantation of KSL HSPC

Long bones, pelvis and spines were harvested and muscle tissue was removed. Bones were crushed in PBS complemented with 2 mM EDTA (Sigma) and 0.5% BSA (Sigma) and bone marrow cells in suspension were filtered on a 40 um cell strainer. Lin- cells were obtained using the Direct Lineage Cell Depletion Kit (Miltenyi Biotec) according to the manufacturer's instructions. Cells were then stained with antibodies against HSC-related markers Kit and Lin- Kit+ Sca+ (KSL) cells were sorted from *Tet2*[fl/fl] Mx-1 Cre mice.

Transduction of mouse HSC was performed in serum free-medium enriched with cytokines as previously described[119] at a MOI of 20. Transduction efficiency was monitored by qPCR. Sixteen hours after transduction cells were washed in PBS and transplanted at a dose of $1 \times 10^4$ cells via retro-orbital injection in lethally irradiated C57BL6/J recipients together with $2 \times 10^5$ Sca-depleted supporter BM cells. Six weeks after transplant, 4 doses of 12.5 μg/g pI:pC (Amersham) were administered intraperitoneally to induce Cre activity (and thus *Tet2* deletion).

Mice were monitored weekly for body weight and signs of suffering, and euthanized when showing ≥ 15% weight loss and/or labored breathing, followed by necropsy analysis. Serial collections of blood from the mouse tail were performed to monitor the hematological parameters and donor cell engraftment. At the end of the experiment (25 weeks), BM and spleen were harvested and analyzed (scATAC-seq, flow cytometry for hematopoietic subpopulations).

## PVA-based HSC cultures and analyses

For PVA-expansion experiments, Lin- Kit+ Sca+ Cd150 + Cd48- Epcr + cells were sorted from *Tet2*[fl/fl] mice. Culture was performed as previously described[73]. After 6 days of expansion, cultures were split, and transduced with *Sox4* OE lentiviral vector or BFP control vector (MOI 40). Four days later, cultures were split again and transduced with Cre-expressing lentiviral vector (MOI 40) to introduce *Tet2* KO. Levels of transduction were monitored by BFP expression and qPCR. KSL cells were enriched again by FACS before proceeding with other analyses (SHARE-seq, flow cytometry for HSC markers, transplant). For transplant, $6 \times 10^4$ cells from each condition were transplanted in lethally irradiated C57B6 recipients together with $2 \times 10^5$ competitor total BM cells (mismatched for CD45 isoform expression). Serial collections of blood from the mouse tail were performed to monitor donor cell engraftment. At 30 weeks after primary transplant, $2 \times 10^6$ cells from each primary mouse were transplanted in secondary lethally irradiated recipients.

For experiments including the high complexity barcoding library LARRY[83], transduction of Lin- Kit+ Sca+ CD150 + CD48- EPCR+ cells was performed 2 h after sorting in a serum free-medium enriched with cytokines[119] at a MOI of 20. Transduction level around ~20% was achieved using these conditions, thus maximizing the likelihood of vector copy number of 1 (1 unique barcode/ cell). Sixteen hours after transduction, cells were washed in PBS and switched to PVA-based medium. Larry lentiviral libraries were prepared from plasmid stocks as described[83], and diversity was confirmed to be in the range of $2 \times 10^5$. Libraries were prepared as described[83], and sequenced on $2 \times 150$ Miseq (Illumina). Clonal abundances were estimated using a pipeline adapted from ref. 120. Briefly, barcodes are isolated by the identification of flanking sequences using the ShortReads R package, and further filtered by perfect matching of the constant bases present within the 28-mer barcode. Correction for sequencing errors is performed using the Starcode algorithm[121] using default parameters. Low-frequency barcodes with counts <10 are removed.

## Flow cytometry and FACS

Immunophenotypic analyses and cell sorting were performed on FACS Aria II (BD Biosciences) and antibodies utilized are listed in Supplementary Data 5 (with corresponding catalog numbers, dilutions utilized and RRID #). Single stained and Fluorescence Minus One stained cells were used as controls. For peripheral blood analysis, red blood cells were lysed using ACK buffer (Quality Biologicals) for 7 minutes at room temperature before staining. Samples were incubated with the antibody cocktail in PBS 2% FBS for 30 minutes at 4 °C before analysis or sorting. 7-AAD Viability Staining Solution (BioLegend) was included in the sample preparation for flow cytometry to exclude dead cells from the analysis. Relevant gating strategies are reported in Supplementary Fig. 9.

## Metabolomics analysis

For metabolomics analysis, at least 3 biological replicates were used for each GMP clones and primary sorted HSC samples from mouse BM. Samples were pelleted and lysed by adding $100\,\mu L$ ice-cold 80% methanol in water and polar metabolites were extracted using a methanol-chloroform phase separation (1 ml methanol containing $2.5\,\mu M$ of an internal standard (fully 13C-, 15N-labeled amino acid mix; Cambridge Isotope Laboratories), $500\,\mu L$ water, and 1 mL chloroform). The samples were mixed on shaker at $4\,°C$ for 15 min and centrifuged at 5000 g 15 min at $4\,°C$. The aqueous phase was recovered and dried under nitrogen flow and resuspended in 30% acetonitrile in water, with a volume scaled to the extracted cell number. $14\,\mu L$ of each sample was transferred to glass microinserts and used for MS1 runs. The rest of each sample was combined to create a pool sample for MS2/MS3 acquisition. Samples were run a Vanquish LC coupled to an ID-X mass spectrometer (Thermo Electron North America, Madison, WI, USA). A volume of $5\,\mu l$ was injected on a Zic-pHILIC column (150×2.1 mm, 5 micron particles; Merck). The flow rate was $0.15\,mL\,min^{-1}$, except for the first 30 s where it was ramped from $0.05\,mL\,min^{-1}$ to $0.15\,mL\,min^{-1}$. The mobiles phase were 20 mM ammonium carbonate in water with 0.1% ammonium hydroxide for A and acetonitrile 97% in water for B. The gradient consisted of an isocratic step of 0.5 min at 93% B, then a gradient to 40% B in 18.5 min, then to 0% B in 9 min, followed by an isocratic step at 0% B for 5 min and back to 93% B in 3 min. The column was re-equilibrated at 93% B for 9 min. For MS1 only run, data was acquired in MS1 full-scan with polarity-switching, resolution 120,000, RF lens 30%, normalized AGC target 25%, max IT 50 ms, m/z range 65 to 1000. For MS2/3 runs (on the pool sample), data was acquired using AquireX deepscan with 5 repetitions, separately in each polarity. For the targeted analysis, a standard mix at $1\,\mu M$ of each target was prepared and run after the samples to confirm retention times. Targeted metabolite measurements were normalized to the internal 13 C/15N-labeled amino acid standard. For untargeted analysis, Compound Discoverer (CD, version 3.3, Thermofisher Scientific) was used to generate a list of compounds (monoisotopic molecular weight and RT couples, de-adducted, combined from positive and negative mode) and to integrate the corresponding area. Identification was based on the MS2/3 data acquired on the pool sample. Fragmentation spectra were searched against an internal library (with matching retention time, to generate level 1 identification) and mzCloud (to generate level 2 and level 3 identifications). All identifications were manually curated. For untargeted metabolomics functional analysis, area and m/z (as [M + H] + adducts calculated from the monoisotopic masses generated in CD) were processed using Metaboanalyst 5.0 software[118]. Positive ion mode, 5.0 ppm tolerance and retention time (minutes) were provided as input parameters. Mouse KEGG database was used for Mummichog Pathway Analysis. Normalized AUC data relative to Fig. 6e and Supplementary Fig. 8e are reported in Supplementary Data 6.

## Statistical analysis

Data were expressed as means ± SEM or dot plots with median values indicated as a line. Statistical tests and number of replicates are reported in the figure legends. Assumptions for the correct application of standard parametric procedures were checked (e.g., normality of the data). Adjusted $p$-values using Bonferroni's correction are reported. Whenever these assumptions were not met, nonparametric statistical tests were performed. In particular, Mann-Whitney test was performed to compare two independent groups. In presence of more than two independent groups, Kruskal-Wallis test was performed, followed by post hoc pairwise comparisons. For paired observations, Wilcoxon matched-pairs signed rank test was performed. For statistics in Fig. 2g, paired $t$-test was used because each independent comparison was performed using *Tet2* KO and WT cells competitively transplanted in a single mouse. Analyses were performed using GraphPad Prism 10 and $R$ statistical software. Differences were considered statistically significant at $*p < 0.05$, $**p < 0.01$, $***p < 0.001$, $****p < 0.0001$, "ns" represents non significance.

## Reporting summary

Further information on research design is available in the Nature Portfolio Reporting Summary linked to this article.

## Data availability

scRNAseq, scATACseq and bulk ATACseq datasets are deposited in the GEO database under accession GSE247970. sc-ATAC-seq and sc-RNA-seq processed datasets can be also explored using a custom interactive application [https://buenrostrolab.shinyapps.io/gmps/]. Metabolomics data are deposited in the MetaboLights database under accession MTBLS9914. Source data are provided with this paper.

## Code availability

The code utilized for data analysis is available at [https://github.com/buenrostrolab/GMP_analyses_code].

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

## Acknowledgements

We thank all the members of Buenrostro and Scadden labs for useful discussion. We also thank David Sykes (MGH) for suggestions related to the Hoxb8 clonal system and Ruslan Soldatov (MSKCC) for critical discussion of the work. We are grateful to the Harvard Stem Cell Institute-Center for Regenerative Medicine Flow Cytometry Core Facility at MGH for technical assistance with FACS sorting, Bauer Core Facility at Harvard University for high-throughput sequencing assistance and Harvard Center for Mass Spectrometry for assistance with metabolomics analyses.

Funding sources to this work include: NIH (5P01HL131477-03), MDS Edward P. Evans Foundation (Discovery Research Grant 2021), Gerald and Darlene Jordan Chair at Harvard Medical School to D.T.S., Gene Regulation Observatory at the Broad Institute of MIT & Harvard, the Chan Zuckerberg Initiative, and the NIH New Innovator Award (DP2) to J.D.B., EMBO long-term fellowship (ALTF-743-2018) and AICF 2018 fellowship to G.S., Swedish Research Council's International Postdoc grant to T.A.K, Walter-Benjamin fellowship from the German Research Foundation (Deutsche Forschungsgemeinschaft [DFG], GZ: MA 9452/1-1) to C.M., Kirschstein National Research Service Award (NRSA) Institutional Research Training Grant T32 program to C.R.

## Author contributions

G.S. conceptualized the project, designed and performed experiments, analyzed data and wrote the manuscript. V.K. analyzed data, performed bioinformatic analyses and wrote the manuscript. F.D., T.K. and C.M. designed, performed experiments and analyzed data. R.S., A.E., Y.H. and T.T. performed experiments. C.R. helped establish the Hoxb8 line. J.B. conceptualized the project, provided supervision, performed bioinformatic analyses and wrote the manuscript. D.T.S. conceptualized the project, provided supervision, and wrote the manuscript. All authors read and approved the final manuscript.

## Competing interests

G.S. is currently an employee of Tessera Therapeutics. D.T.S. is a founder, director and stockholder of Magenta Therapeutics, Clear Creek Bio, and LifeVaultBio. He is a director of Agios Pharmaceuticals, Editas Medicine and Sonata Therapeutics and a founder and stockholder of Fate Therapeutics and Garuda Therapeutics. He is a consultant for VCanBio and SAB member of Simcere Pharmaceuticals. J.D.B. is a SAB member of Camp4 and seqWell. Other authors declare no competing interests.
