## [Peer Review File · Nature Communications]

Cell of origin epigenetic priming determines susceptibility to Tet2 mutationReviewers' Comments:

Reviewer #1:

Remarks to the Author:

In the manuscript titled "Cell of origin epigenetic priming determines susceptibility to Tet2 oncogenic mutation", Schirotti et al address epigenetic priming as a key factor in the deleterious effect of TET2 mutation in murine HSPC and GMP cells. They have shown shown that priming by SOX4 in HSCs and GMP leads to enhanced effects of TET2 KO in the same cells. They took advantage of scATAC and scRNA-seq to address this and using DORC method they identified heterogeneity in GMP populations corresponding to various gene activities. They narrowed down to SOX4 and showed that priming HSCPCs with SOX4 even before TET2 KO enhances the functions of TET2. The paper present a wealth of different techniques and perspectives with interesting models for epigenetic priming.

Please find below areas where the work may be enhanced to improve readability and better support claims.

1. The authors claim in the title of their work that the epigenetic state of the cell of origin defines the sensitivity to Tet2 mutation. It is not entirely clear if the premise of this work is the evaluation of the effects of the somatic mutation at different differentiation stages, or heterogeneous epigenetic states of the same cell type. In addition, when referring to cell of origin, the authors focus the writing on hematopoietic stem cells (HSC) which are the cells in which the initial somatic mutation originates, thus supporting the clonal expansions, while the bulk of the work is performed on the GMP population.
2. While the initial characterization of GMP stages in vivo is of interest, novelty is unclear. Previous studies (including those cited by the authors here) have thoroughly defined the heterogeneity of the GMP compartment at both transcriptional and chromatin accessibility levels. While this is an important characterization of their system, perhaps the authors could clarify what new findings are presented in Fig.1 and what is the link to downstream analyses.
3. Although the spread in the distribution of single cell scores for accessibility of Tet2 associated peaks is interesting, it seems to be restricted to upregulated peaks in the Tet2 KO mice, rather than the peaks that are downregulated in Fig. 2B. In addition, since GMP populations were pooled for the differential analysis, this can arise from population frequencies rather than effects of the mutation within a cell type or cellular state.
The authors should perform differential analysis within cluster (GMP-stem, GMP-Neutro or GMP-mono) and verify that the increase in dispersion is not an artifact of changes in cellular frequencies observed due to the Tet2 loss.
4. The authors highlight the CTCF binding motif as enriched in downregulated scATAC peaks, and speculate that Tet2 KO-induced hypermethylation can be disrupting CTCF binding and thus chromatin conformation. With this premise, how do the authors explain the enrichment in CTCF motifs also in the upregulated (more accessible) peaks in the Tet2 KO mice?
5. It is not clear why the paired clone analysis was performed in the presence of GM-CSF or SCF, since these will result in the convergence of the GMPs towards the same initial epigenetic state. Since the premise of the paper is that epigenetic heterogeneity drives differential response to loss of Tet2, why do the authors create a homogeneous epigenetic state before inducing the mutation? Also, it is not clear why the number of clones is so unbalanced between SCF (n = 4) and GM-CSF (n = 24). As the authors mention, GM-CSF and SCF result in distinct differentiation stages of the GMPs, towards a more mature granulocyte-monocyte state or to an earlier granulocyte-primed state, respectively. Thus, the epigenetic differences between these stages are more likely derived from the differentiation process rather than from heterogeneity in the cell population. Therefore, the conclusions regarding influence of epigenetic state are confounded by the differentiation process occurring upon addition of GM-CSF or SCF. Have the authors verified the heterogeneity between clones beyond the PCA space? Applying additional measurements of heterogeneity such as Shannon index could be of use. Additional validation is suggested to define that the epigenetic states are indeed heterogeneous and distinct between clones.
- 6.

7. Figure 3a may benefit from additional detail. For example, it is not clear how TET KO is induced in the illustration.
8. Figure 3c. Did the author see any differences in the accessibility of IL-6 or other inflammatory cytokines that can be correlated with the flow data of the clones?
9. Figure 3e – It is a bit hard to understand what is being shown. Are these only SCF treated cells? what does the colors above the heat map denote?
 - Figure 4d – The color code of WT clone is not clear, are these cells treated with GM-CSF/ SCF or is it a comparison between WT and KO cells. What is WT clone # and why is important here? What is the color scale for the Functional Perturbation score?
10. There is insufficient detail and clarity about the clonal assay and how the score computed.
 - Why did the author focus on SOX4 alone? It is not clear why the authors decided to focus on Sox4, since multiple TFs were associated with the functional response. In addition, how frequent are Sox4 mutations / overexpression present in clonal hematopoiesis or other malignancies? Did the authors try other genes perturbation and see if the changes observed are unique to SOX4? For example, KD of genes that are negatively correlated can be performed to strengthen the data (especially in low TET2 KO sensitive clones, inhibiting them will increase sensitivity). 4H shows that by Day 6, most of the effect is TET2, with limited impact of SOX4 OE. Also 4h shows in vivo seems to be additive. Perhaps not surprising given that SOX4 has known effects on HSPCs. How can the cooperativity be shown more conclusively? Overall, the Sox4 overexpression seems to enrich in earlier stem cell states, which are known to be more susceptible to Tet2 mutations. While interesting, these findings do not directly support the main claim of the manuscript regarding epigenetic heterogeneity driving differential functional outcomes to loss of Tet2 function.
11. More details needed on how functional perturbation score is normalized and calculated.
12. Does SOX4 controls the differentiation of Neutrophils? Is there any molecular mechanism on why SOX4 OE TET KO cells accumulate MPP4 cells and decreased mature differentiation?
13. What is the status of lymphoid cells in the mice transplanted with SOX4 OE TET2 KD cells, was there any changes in the percentage of the lymphoid?
14. Mechanisms on how SOX4 can be activated or epigenetic priming in GMPs needs to be commented on.
15. The authors should strongly consider depositing the code in a public repository for reproducibility.
16. Additional mice in the control group of Tet2 fl/fl should be considered for the initial experiment (Fig. 1). Since authors state there are changes in cell frequency, statistical analysis supporting this claim are needed to estimate the reproducibility across biological replicates.
17. The cluster name "Mono-biased GMP" in Fig 1B is assigned to two different clusters. What are the differences between these two? Were they pooled for downstream analysis?
18. Labeling of the top annotation in the heatmap of Fig. 1D is required as well as color encoding. Color scale for each feature should be shown.
19. The authors show the residuals between DORC and RNA expression for Klf4 in particular (Fig. S1E). However, these dynamics are not show for control and Tet2 -/- separately, which seems like a missed opportunity. Do the dynamics differ between genotypes? Can this be associated with the monocyte progenitor skewing?
20. The authors should consider adding orthogonal validation of the cell cluster identity using reference mapping to verify the manually defined cell cluster labels.
21. Claims such as "we observed differential localization across different WT clones (FigS4C)" are not clear. What is the definition of "differential localization"? How was it calculated? What statistics were used? A clearer language or statistical support is required for these claims.

22. In Figure 2C, the x axis labels are not clear. What is observed and what is expected? Is this Tet2 KO – WT? A clearer label should be used, and details should be included in the figure legend.
23. In some figures the axis labels are overlapping with the numbers indicating the values of the variable.
24. In Figure 4A, it is not clear why the WT column is necessary since it has been normalized to 1 for all clones. Consider showing the dispersion plotting Tet2 KO to WT Euclidean distance directly.
25. In Figure 4F, the authors show “% of localization”. This metric is very obscure and not explicitly detailed in the main text nor the methods section. It is hard to interpret the results without that information.
26. Figure 5E: It is not clear the metric that is being represented here. Also, the color legend states “log FDR”. Assuming this is $-\log_{10}(\text{FDR})$, obtaining a value of 300 indicates that the statistic is not well calibrated.
27. The reasoning for performing metabolomic analysis is not entirely clear, and seems somewhat disconnect from the rest of the work.

Reviewer #2:

Remarks to the Author:

Clonal hematopoiesis (CH) is a widely studied age-related phenomenon associated with increased overall mortality due to elevated risks for haematologic malignancies and cardiovascular diseases. One of the most frequently mutated genes in CH is TET2. A poorly understood aspect of CH is its incomplete penetrance, with most patients remaining ostensibly disease-free, suggesting additional alterations are required for ostensible pathology to manifest itself. The manuscript by Schirolli et al. investigate in a knockout Tet2 mouse model how different cell states/cell of origins are functionally impacted by the loss of Tet2. The authors conduct single cell ATAC-seq and RNA-seq profiling of the murine bone marrow progenitor compartment to identify genomic alterations upon the loss of Tet2. An in vitro immortalized GMP model is then used to study genomic and functional differences upon Tet2 loss in clonal pairs of WT and KO clones, in conjunction with various functional analyses (e.g., phagocytosis, ROS production). The authors further identify Sox4 as an interesting candidate to cooperatively be responsible for the effects of Tet2 loss (e.g., Sox4 enhances the epigenetic dysregulation observed after Tet2 KO).

The manuscript is well-written, of overall very good quality, and of direct relevance to our understanding of CH. However, several aspects appear only touched upon in a rather superficial manner, leaving various loose ends. Moreover, the comprehension and readability of the work could be improved further by providing important details as outlined below. A more direct discussion of the relevance of their findings to CH would further be very valuable. This reviewer has the following major comments:

1. Another depiction of the skew in lineage trajectories in Fig. 1F should be provided. Is this skew consistently reproduced across the different mice that were used? Potentially bar plots resolved by cell state would facilitate such a quantification.
2. What's the degree of splenomegaly? Fig. S2B only provides a binary and no quantitative readout.
3. The authors describe a variety of findings but show no matching data, which needs to be corrected throughout. Examples include:
 - “Efficient Tet2 deletion was confirmed at the genomic level by ddPCR in >95% of circulating cells.”

- "We also consistently observed a decrease in Ctf motifs, which are sensitive to DNA methylation."

4. Along these lines, basic quality control data should be shown for all single cell genomic datasets in the study. How many cells were profiled, pass QC, number of cells per mice, etc. Is data quality comparable, or is the signal driven by or more pronounced in 1-2 mice, given biological heterogeneity etc?

5. Regarding the analysis in Fig. 2A and the following, which GMP populations were compared? As four GMP subpopulations were identified, with two belonging to a GMP-Mono state, this analysis should be conducted in a cluster-specific manner, in particular given some potential skew in the frequency of these populations as suggested in Fig. 1F.

6. Along these lines, this reviewer finds it surprising that differential up- and down-regulated peaks are so distinctly distributed across the chromatin landscapes of these cells (Fig. 2D). Do the authors have an explanation for this phenomenon? Are some of these differential changes already discernible in the respective later or earlier stages? Or phrased another way, are the differentially downregulated peaks a consequence of the previously upregulated peaks (resolving this relationship would go a long way in this reviewer's opinion)? Is there any correlation with Tet2 expression levels across the landscape (in the wildtype setting)? Can the authors show examples of differential chromatin peaks in track plots to give the reader a better intuition of these findings?

7. Why was no matching analysis to Fig. 2A conducted for the HSPC compartment? As in particular the GMP-stem signature extends to this compartment (Fig. 1E), and many of the differential peaks are also identified in HSPCs (Fig. 2D) it appears more than worthwhile to provide a comprehensive analysis across the trajectory of monocytic differentiation, be the major focus the GMP compartment.

8. Regarding Fig. 2G, what does ROS production at baseline (-PMA) look like? Is there already a difference between WT and Tet2 KO? Also, the statement "Mature Tet2 KO myeloid cells from both lineages showed increased reactive oxygen species (ROS) production upon treatment with PMA (Fig.2G). This finding is consistent with a described role for Tet2 in repressing inflammatory gene expression." needs to be clarified. How is ROS production here linked to alterations in gene expression?

9. About Fig. 3B: How do WT and Tet2 KO clones cluster in 3B? Presumably, the pairs cluster closely together, or are the alterations more obvious than in the prior data?

10. Data in Fig. S3D-G should be summarized, showing mean and standard deviations. The 29 pairs of bar plots should be kept in the supplement, but these alone are challenging to digest. Can the authors show chromatin track plots for the differential loci, e.g., IL6, Cd115, Ly6G.

11. About the subsequently introduced functional perturbation score. Can the authors provide more details on the methods on how exactly this has been calculated? What's the major driver of that score? Also please highlight the GM-CSF clones in Fig. 4B, or were these not transplanted? Does the in vivo output correlate with the functional perturbation score?

12. The authors nominate Sox4 as an interesting gene candidate from their GMP clone data. Do the authors also see any indication of Sox4 activity/alteration in their Fig. 1-2 data? Please show flow plots for the Fig. 4H data (presumably flow cytometry was used here), as differences in CD117 expression appear rather subtle and the bar plots do not account for the distribution of the signal. Can the authors comment on how Sox4 and Tet2 may cooperate at the chromatin level? Do they bind to distinct or the same target genes? Is the Sox4 motif and as such target gene regulatory elements known?

12. Please show an annotated UMAP of the dscATAC-seq data discussed in Fig. 5. While a UMAP is

shown in 5F, its unclear whether this relates to data in Fig. 1A, or whether the new data was projected onto the same landscape, etc. As noted above, basic QC metrics should be shown, alongside how consistent the results are across the mice/replicates used. Analogous to the previous point 5, which GMP (sub)population(s) was analyzed?

13. The authors should add a discussion about how directly their findings related to the CH in a human setting. One would expect that TET2 mutations will ultimately arise only in a self-renewing population, presumably HSCs. Would the authors therefore suggest that subtle differences between HSCs, predetermine long-term consequences, or that such a self-renewing population may further diversify and then additional subclones to become more susceptible to developing pathology? In particular, as the CH population may evolve over many years if not decades.

14. The GEO data is not accessible. A reviewer access token should be provided.

Minor comments

1. Why did the number of GM-SCF (4) and SCF (24) derived clones used for analysis differ?

2. Fig. S3C: can the authors show pair-wise flow plots of the different markers? Based on the histograms alone, the phenotypic differences in marker expression are difficult to compare.

3. Fig. S3D-G: please indicate the statistical tests used.

4. Revise the use/introduction of abbreviations (should be introduced when used first, e.g., dsc, OE, pI:pC)

5. Details in the materials and methods need to be revised for consistency (uM -> μ M, C -> °C, 105 - >10⁵/10E5, % -> f %(v/v), agitated (rpm?) 2⁻DCt instead of 2^ΔCT, 30[^]vs 30 min), etc.

Reviewer #3:

Remarks to the Author:

In this very complex manuscript, the authors comprehensively examined the effects of TE2 deletion in different hematopoietic stem and progenitor populations in a conditional TET2 KO mouse model. They describe the epigenetic landscape of TET2 mutated hematopoiesis in detail and provide evidence that the epigenetic state of the cell of origin also determines the phenotype of the cell when TET2 is mutated. The authors then go on to identify SOX4 as a driver of increased myeloid output in their system by using an elegant system of immortalised myeloid progenitor cells enabling single cell expansion and exact timing of differentiation by stimulation with either SCF or GM-CSF in vitro. TET2 KO cells showed distinct chromatin accessibility patterns depending on stimulus towards neutrophil or myeloid-monocyte differentiation.

The manuscript contains extensive datasets and the authors are to be commended for their thorough characterisation of the epigenetic effects of TET2 deletion on hematopoietic subsets, which provides a valuable reference for other groups. However, the results section (especially pertaining to Figures 1 and 2) are at times quite difficult to follow, especially since the description of the results in figure 2 includes many references to the current literature without necessarily following up on this data in the rest of the manuscript. I think the results section would benefit from streamlining to increase readability and clarity without extensive discussion of observations that do not play a role in the rest of the story. The introduction and discussion in contrast are very clear and well written.

Figure 1F it seems that TET2 KO cells are also skewed towards an MEP phenotype but this is not recapitulated in colony data in Fig S2C. Can the authors comment.

Fig 2B peak distribution WT vs TET2 KO (in black) is barely readable.

Fig 2C shows repressed peaks in chromatin regions for transcription factors that are involved in regulation of monocyte commitment: this is perhaps surprising as in the clinic one of major phenotypes of TET2 mutation is usually monocytosis. Can the authors discuss.

Fig 2G recently published work has shown that TET2 mutated neutrophils differentiated from human HSPC in a NSG transplant model have defective phagocytosis (Huerga Encase et al, Cell Stem Cell 2023). So this observation is relevant, but not novel. This work should be cited in this context.

While the mouse model nicely recapitulates the myeloid bias in TET2 mutated CH, previous work (Buscarlet 2018) has shown that in human CH TET2 arises in a later cell of origin than for example DNMT3A, i.e. committed HSC vs a multipotent HSC. How does this compare to the mouse data here. Could the occurrence of TET2 mutations lower in the human hematopoietic hierarchy explain the GMP heterogeneity seen in the mouse model?

Cell of origin epigenetic priming determines susceptibility to *Tet2* mutation

We thank the Reviewers for the careful evaluation of our manuscript. We have substantially revised the work to include the suggestions and to address all the concerns raised. Please find a detailed point-by-point discussion below.

Point-by-point reply to the Editor and Reviewers

Referees' comments:

Reviewer #1 (Remarks to the Author):

In the manuscript titled "Cell of origin epigenetic priming determines susceptibility to *Tet2* oncogenic mutation", Schirolli et al address epigenetic priming as a key factor in the deleterious effect of *TET2* mutation in murine HSPC and GMP cells. They have shown that priming by *SOX4* in HSCs and GMP leads to enhanced effects of *TET2* KO in the same cells. They took advantage of scATAC and scRNA-seq to address this and using DORC method they identified heterogeneity in GMP populations corresponding to various gene activities. They narrowed down to *SOX4* and showed that priming HSCPCs with *SOX4* even before *TET2* KO enhances the functions of *TET2*. The paper presents a wealth of different techniques and perspectives with interesting models for epigenetic priming.

Please find below areas where the work may be enhanced to improve readability and better support claims.

1. The authors claim in the title of their work that the epigenetic state of the cell of origin defines the sensitivity to *Tet2* mutation. It is not entirely clear if the premise of this work is the evaluation of the effects of the somatic mutation at different differentiation stages, or heterogeneous epigenetic states of the same cell type. In addition, when referring to cell of origin, the authors focus the writing on hematopoietic stem cells (HSC) which are the cells in which the initial somatic mutation originates, thus supporting the clonal expansions, while the bulk of the work is performed on the GMP population.

We thank the Reviewer for the constructive suggestion. Our goal was to define whether epigenetic features within the same cell type affect the phenotype of a specific mutation and to determine if the state of differentiation affected the phenotype. We used a GMP model but validated our findings in HSC and try to present our work accurately balancing the differences between model, validation and disease. Our abstract is now modified to better articulate these issues as follows:

*Hematopoietic stem cell (HSC) mutations can result in clonal hematopoiesis (CH) but the clinical outcomes are heterogeneous. While the founder and secondary mutations likely drive emergent neoplastic disease, we hypothesized that the state of the cell in which the mutation occurs impacts the pre-malignant phenotype. Here, we investigated how the cell state before the *Tet2* mutation occurs affects susceptibility to that commonly occurring CH mutation. Using an*

inducible system for myeloid progenitor clonal expansion, we demonstrated that epigenetic features of the cells at a similar stage of differentiation were highly heterogeneous and differentially responded to Tet2 mutation. Further, the differentiation stage of the cells also altered their response to Tet2 mutation. Therefore, the epigenome of the cell of origin modulates clone-specific behavior following acquisition of the Tet2 CH mutation. Evaluating molecular features associated with higher risk functional outcomes, Sox4 fostered a global cell state of high sensitization towards Tet2 KO. Using GMP and primary HSC models, we show that Sox4 promotes cell dedifferentiation, alters cell metabolism and increases the in vivo clonal output of mutant cells. Our results validate the hypothesis that epigenetic features can predispose specific clones for dominance and explain why an identical mutation can result in different phenotypes.

Moreover, we also provided a paragraph in the discussion to highlight the potential limitations of our work:

Limitations of the study. *A potential limitation of our study is the use of an in vitro clonal system for the identification of Tet2-related functions. This model may be suboptimal mainly because i) cells are conditionally immortalized using HoxB8, which could influence chromatin accessibility and alter regions bound by Tet2 and ii) cells are arrested at the GMP state, therefore missing information regarding more primitive progenitors relevant for disease progression. Complementary in vivo studies using clonally tracked HSC-based models such as recently described²⁷ could allow uncovering of other molecular players that determine clonal advantage in CH models. Moreover, a clear distinction between phenotypic effects driven by differentiation stages or heterogeneous epigenetic states within the same cell type can be difficult to determine as these two cellular properties are closely related.*

2. While the initial characterization of GMP stages in vivo is of interest, novelty is unclear. Previous studies (including those cited by the authors here) have thoroughly defined the heterogeneity of the GMP compartment at both transcriptional and chromatin accessibility levels. While this is an important characterization of their system, perhaps the authors could clarify what new findings are presented in Fig.1 and what is the link to downstream analyses.

We thank the Reviewer for raising this issue.

To the best of our knowledge, previous studies extensively characterized the heterogeneity within the GMP compartment by performing unbiased analyses of the transcriptome (some examples: Paul, Cell 2015; Olsson, Nature 2016, Yanez, Immunity 2017, Kwok, Immunity 2020). However, chromatin regulatory activity was in previous literature inferred by expression of known transcription factors within transcriptionally-defined cell states. Single gene regulatory networks have been analyzed in GMP using bulk ATAC-seq or Chip-seq, but an unbiased evaluation remains missing.

Here, we take a complementary approach and characterize the heterogeneity of GMP states using scATACseq. We dissect which chromatin regions are functionally linked with downstream gene expression by independently using domains of regulatory chromatin (DORCs) and

chromatin signatures. Our ATAC-based data defines myeloid cell states which are consistent with prior knowledge (which validates our findings) but also provides a definition of the epigenetic elements that contribute to myeloid identity regulation. We think the combined datasets are unique.

In addition, we also clarified in the text how the GMP epigenetic signatures identified in Figure 1 are then utilized in the further manuscript as reference for mapping the levels of Tet2-mediated epigenetic and functional distortion in both primary cells and GMP clones.

We revised the text as following to highlight these concepts:

Taken together, our unbiased analysis defined epigenetic programs relevant for defining functional identities of primary GMP cells, which are largely consistent with hierarchical differentiation models previously defined by transcriptomic analyses^{43,44}. These GMP signatures represent relevant epigenetic functional states and serve as a blueprint to further map Tet2-mediated distortion of GMP states.

We also now additionally provide complete lists of TF motifs, ATAC gene scores and DORCs that characterize our epigenetic regulatory GMP signatures as an additional resource (new Table S2).

3. Although the spread in the distribution of single cell scores for accessibility of Tet2 associated peaks is interesting, it seems to be restricted to upregulated peaks in the Tet2 KO mice, rather than the peaks that are downregulated in Fig. 2B. In addition, since GMP populations were pooled for the differential analysis, this can arise from population frequencies rather than effects of the mutation within a cell type or cellular state.

The authors should perform differential analysis within cluster (GMP-stem, GMP-Neutro or GMP-mono) and verify that the increase in dispersion is not an artifact of changes in cellular frequencies observed due to the Tet2 loss.

We agree with the reviewer that the spread distribution for single cell scores is more prominent for upregulated peaks, and we changed the text as following to highlight this point:

In order to evaluate the level of epigenetic dysregulation on a single cell basis, we calculated single cell scores for accessibility of Tet2 differential peaks from Fig.2A (Fig.2B). We observed a high spread in the distribution of Tet2 KO cells as compared to WT (especially for peaks upregulated in Tet2 KO), suggesting that there is a level of heterogeneity in how single GMP cells are epigenetically affected by Tet2 mutation.

In order to address the Reviewer's concern regarding pooled GMP analysis, we performed several new analyses and highlighted some clarifications:

i) we first generated two new panels (Figure S1J-I), which collectively show the distribution of cells across the different clusters. Apart from some batch-effect related changes in the proportion of cells belonging to differentiated clusters, we don't detect significant skewing in the GMP-specific clusters (1,4,7,10,13). We further clarified this concept in the text and highlighted that UMAP from Figure 1F suggests not a redistribution of cells across clusters but rather small changes in states as following:

Finally, we analyzed how *Tet2* mutant cells are localized across the differentiation trajectory. In accordance with results from previous RNA-seq studies³², we observed that *Tet2* knockout does not result in independent cell clusters, but rather results in the redistribution of cells within selected neighborhoods (**Fig.1F**). This suggests that *Tet2* mutation perturbs the chromatin state of primitive hematopoietic progenitors, but effects on cell identity are minimal since no significant skewing in cell abundances across clusters is observed (**Fig.S1J-K**).

ii) We clarified in the text that the differential analysis performed on pooled GMP was normalized by cluster identity as a covariate (as indicated in the corresponding methods section). We further highlighted in the legends of Figure 1 and 5 and methods section which populations were considered for the differential analysis.

iii) Although we agree with the Reviewer that cluster-specific comparison would provide useful insights, we provide a new panel (Figure S3A) where we model the scenario when differential analysis is performed by individual cluster. By repeating the differential analysis in bulk GMPs or using only smaller fractions of cells in the dataset, we show that our differential analysis is underpowered when analyzing small cell numbers. We believe that increasing the sensitivity of detection of *Tet2* differential peaks is really valuable in this context.

iv) We added DORC differential analysis performed across GMP clusters, showing extensive alterations at the level of regulators of myeloid differentiation (new figure Fig.S3C, new Table 3).

4. The authors highlight the CTCF binding motif as enriched in downregulated scATAC peaks, and speculate that *Tet2* KO-induced hypermethylation can be disrupting CTCF binding and thus chromatin conformation. With this premise, how do the authors explain the enrichment in CTCF motifs also in the upregulated (more accessible) peaks in the *Tet2* KO mice?

We thank the Reviewer for raising this important point. Consistent with its catalytic function as a DNA demethylase, a direct role for *Tet2* in maintaining open chromatin and allowing TF binding in hematopoietic cells (and thus, in its absence, hypermethylation of DNA can impede binding to the DNA of transcription factors including *Ctcf*) is well established. On the other hand, less is known about increases in chromatin accessibility observed after *Tet2* KO. We hypothesize that the increases in chromatin accessibility observed here may be due to accumulated epigenetic dysregulation or associated secondary events in response to the absence of *Tet2*. As supporting evidence for this hypothesis, previous work described a poor co-localization of *Tet2* in regions with increased accessibility after the KO in contrast to the regions with reduced accessibility (Rasmussen, Genome Res 2019, *Tet2* ChIP-seq performed in myeloid progenitors immortalized by AML1-ETO).

In addition, increased CTCF occupancy was specifically observed by others in primary AML samples bearing *TET2* mutations (Muhajed et al, Blood 2020). In this context, it was reported that *Tet2*-specific methylation occurs outside of CTCF-binding sites and CTCF increased accessibility overlaps with sites lacking CpGs, making them insensitive to DNA methylation.

5. It is not clear why the paired clone analysis was performed in the presence of GM-CSF or SCF, since these will result in the convergence of the GMPs towards the same initial epigenetic state. Since the premise of the paper is that epigenetic heterogeneity drives differential response to loss of *Tet2*, why do the authors create a homogeneous epigenetic state before inducing the mutation? Also, it is not clear why the number of clones is so unbalanced between SCF ($n = 4$) and GM-CSF ($n = 24$). As the authors mention, GM-CSF and SCF result in distinct differentiation stages of the GMPs, towards a more mature granulocyte-monocyte state or to an

earlier granulocyte-primed state, respectively. Thus, the epigenetic differences between these stages are more likely derived from the differentiation process rather than from heterogeneity in the cell population. Therefore, the conclusions regarding influence of epigenetic state are confounded by the differentiation process occurring upon addition of GM-CSF or SCF. Have the authors verified the heterogeneity between clones beyond the PCA space? Applying additional measurements of heterogeneity such as Shannon index could be of use. Additional validation is suggested to define that the epigenetic states are indeed heterogeneous and distinct between clones.

We apologize for the lack of clarity on this important concept.

As correctly stated by the Reviewer, the focus of our work is to study how epigenetic heterogeneity drives differential phenotypic and molecular response to Tet2 KO. Epigenetic heterogeneity can be described across two independent cell properties: clonal state within the same cell type and differentiation state (differences between cell types).

Although it is difficult to completely separate the concepts of clonal state and differentiation state, comparison across different SCF clones can be considered as a measure of clonal state, whereas comparison between SCF and GM-CSF clones can inform on the effect of differentiation state. Specifically, we utilized a high number of SCF clones to study how epigenetic heterogeneity within the same clonal state affects functional response after induction of Tet2 inactivating mutation (comparison within SCF clones). We believe this is the most relevant and interesting question, because it could provide some mechanistic insight on how clonal heterogeneity could explain the phenomenon of incomplete penetrance seen in humans (Please also see minor comment 1 from Reviewer 2).

We acknowledge that many of the combined GM-CSF+SCF clonal readouts might be misleading and, as suggested, revised/added the following panels.

i) We highlighted the cell type ID (SCF vs GM-CSF) in Figure 3C, new Figure 4A, Figure 4B,C, Figure 4D, new Figure 4E in every panel where the two different cell types are analyzed together. Overall, all these panels highlight a marked distinction between SCF and GM-CSF clones, suggesting that differentiation state plays a role in determining the response to the Tet2 mutation. On the other hand, all these combined analyses also highlight that only a subset of SCF clones are functionally sensitized.

ii) We added a new panel (Figure 4F), to demonstrate that the differentiation state is not the only factor impacting response to Tet2. We correlated chromatin profiles of WT clones to the GMP signatures identified in Figure 1 (stem-like, neutrophilic, monocytic). Differentiated GM-CSF clones showed high monocytic/neutrophilic and low stem-like signatures as expected. Focusing within the SCF clones, we did not find a good correlation between sensitized clones and a GMP stem-like signature, suggesting that other clonal features rather than just differentiation state contribute in predicting sensitivity to Tet2 mutation.

iii) We revised the text of Figure 4 (please see text changes highlighted in red) to clarify that comparison between SCF and GM-CSF groups allows evaluating the effect of Tet2 KO between different differentiation states, whereas comparison within SCF clones (n=24) allows comparison of clonal states within the same cell type.

iv) We revised the text of Figure 4 to highlight that the SHARE-seq data from Figure 4 G-H (former Fig.4E-F) was performed only using SCF clones to further demonstrate the concept of clone-specific epigenetic programs (such as the one mediated by Sox4) as predictors of functional behavior. Indeed this analysis shows at the single cell level that highly sensitized clones are enriched in a separate region compared with non sensitized clones.

In addition, to address the clonal heterogeneity point (“why do the authors create an homogeneous epigenetic state before inducing the mutation” and “additional measurements of heterogeneity are needed to verify that epigenetic states are indeed heterogeneous”), we now provide a more detailed analysis aimed at selectively analyzing the epigenetic state across GM-CSF (n=4) and SCF clones (n=24):

i) We added a new panel (Figure S4E) representing Pearson correlation of epigenetic states using TF motif scores as features. All analyzed clones are represented (SCF and GM-CSF, WT and Tet2 KO). This analysis shows how heterogeneous the chromatin landscape is across different GMP clones, and highlights that this phenomenon is especially accentuated when comparison is made within SCF clones.

ii) We also provided a second new panel (Figure S4F), which shows TF motifs associated with variability in chromatin accessibility between individual SCF-derived clones. This analysis highlights which specific chromatin features contribute to the heterogeneity observed within SCF clones.

iii) Furthermore, functional validation that SCF clones represent a heterogeneous state has been recently published by our group during this revision (Rhee, Blood 2023), now referenced in Figure 3. This work further supports the concept that GMP clones derived under similar conditions are epigenetically and functionally very heterogeneous, and specific differences in the epigenetic state of progenitors can dictate the type of downstream myeloid cells generated.

7. Figure 3a may benefit from additional detail. For example, it is not clear how TET KO is induced in the illustration.

We thank the Reviewer and have addressed this comment using a revised version of Figure 3A incorporating additional details on the experimental workflow.

8. Figure 3c. Did the author see any differences in the accessibility of IL-6 or other inflammatory cytokines that can be correlated with the flow data of the clones?

We generated a new panel (Figure S5E) which shows track plots for some representative WT GMP clones for differential loci suggested by functional assays from Figure 3C (which were performed after GMP differentiation in mature myeloid cells). We selected markers of differentiation, genes involved in inflammation and genes controlling ROS production. This novel data provided:

i) correlation between chromatin status at the level of GMPs and functional measurements in downstream effector cells, reinforcing the concept that epigenetic heterogeneity at the level of progenitors can be used to predict functional outcomes.

ii) further reinforcement of the concept of heterogeneity within SCF-derived clones

9. Figure 3e – It is a bit hard to understand what is being shown. Are these only SCF treated cells? what does the colors above the heat map denote?

We apologize for the lack of clarity. We added a sentence in the text to highlight that in former Figures 3D-E (now Figures 3 E-F) we utilized both GM-CSF clones and SCF clones as the purpose of these analyses is to broadly link functional properties and epigenetic states as following:

We next sought to determine whether these functional alterations may be explained by heritability of chromatin states at the progenitor level. We collectively analyzed WT SCF-derived and GM-CSF derived clones to capture major components of heterogeneity.

We further amended the heatmap in former Figure 3E (now Figure 3F) to include a description of X and Y axes (variable TF motifs and scores by functional properties, respectively). Color coding on top of the heatmap was removed as we agree can be confusing.

- Figure 4d – The color code of WT clone is not clear, are these cells treated with GM-CSF/ SCF or is it a comparison between WT and KO cells. What is WT clone # and why is important here? What is the color scale for the Functional Perturbation score?

We again apologize for lack of clarity and for overlooking these details. We have revised Figure 4 (please also see comment 11 from Reviewer 2) to improve overall readability and comprehension of the functional differences captured by the Functional Perturbation score for each clone pair.

We added the missing color scales and now color-coded the cell treatment (GM-CSF vs SCF) in the heatmap.

We also clarified the clone IDs by removing color coding and adding numbers (which now allows better comparison of these data with all the other panels in Figure 3, Figure S5 and Figure 4), and revised the figure legend to clarify that analysis from former Figure 4D (now new Figure 4E) performs a genome-wide correlation between DORCs identified and ATAC profiles of WT clones, ranked by their level of functional perturbation after Tet2 mutation.

10. There is insufficient detail and clarity about the clonal assay and how the score computed

To provide more details regarding how the Functional Perturbation Score relates to the clonal assays, we have added a new panel (Figure 4A). This new panel shows the fold perturbation in each measured *in vitro* functional property for each individual clone pair (Tet2 -WT). A single score is then computed. We also amended the methods section (as in subsequent comment 11) to provide more details on the mathematical calculation performed.

- Why did the author focus on SOX4 alone? It is not clear why the authors decided to focus on Sox4, since multiple TFs were associated with the functional response. In addition, how frequent are Sox4 mutations / overexpression present in clonal hematopoiesis or other malignancies? Did the authors try other genes perturbation and see if the changes observed are unique to SOX4? For example, KD of genes that are negatively correlated can be performed to strengthen the data (especially in low TET2 KO sensitive clones, inhibiting them will increase sensitivity). 4H shows that by Day 6, most of the effect is TET2, with limited impact of SOX4 OE. Also 4h shows *in vivo* seems to be additive. Perhaps not surprising given that SOX4 has known effects on HSPCs. How can the cooperativity be shown more conclusively? Overall, the Sox4 overexpression seems to enrich in earlier stem cell states, which are known to be more

susceptible to Tet2 mutations. While interesting, these findings do not directly support the main claim of the manuscript regarding epigenetic heterogeneity driving differential functional outcomes to loss of Tet2 function.

We thank the Reviewer for raising this important point.

Our analyses indeed identified chromatin accessibility in the regulatory regions belonging to several genes correlated with a cell state of high or low functional hypersensitization to Tet2 mutation. In general, these loci can be either drivers of phenotype or regions correlated to a particular differentiation state.

For this work we focused on transcription factors to maximize the likelihood of these candidates to be actual drivers of phenotype, thanks to their pleiotropic mechanism of action.

For this reason, we expect that alteration of the expression of many of the non TF-genes won't alter the sensitization status of the clones (and, accordingly, as correctly pointed out by this Reviewer in question 5, many of the loci seem correlated with a GM-CSF vs SCF differentiation state).

Among the positively correlated transcription factors we focused on Sox4 because:

i) It has not been reported to be either a direct interactor with Tet2 or a direct target. In both of these instances, we reasoned that reduced expression (and not increased expression) is predicted to increase Tet2-mediated dysregulation.

To exert its function, Tet2 is reported to interact with pioneering TFs, whose binding to the chromatin tethers Tet2 itself to the DNA and facilitates the recruitment of additional regulatory elements. For example, Gfi1b (PU1) is reported to directly interact with Tet2 in hematopoiesis (specifically in B-lymphopoiesis, Lio et al., Elife 2016) and its depletion cooperates with Tet2 to induce malignant hematopoiesis (Aivaloti et al., Blood Cancer Discov 2022) by amplifying its epigenetic perturbation.

On the other hand, Gata2 expression is Tet2-dependent, as *Gata2* was downregulated in various *Tet2* knockout settings and that forced expression of *Gata2* decreased the competitiveness of both normal and malignant TET2-deficient cells (Ito, Cell Rep 2019, Shih Cancer Cell 2015)

ii) CEBPa, direct interactor of Tet2 (Sardina, Cell Stem Cell 2018), was reported to directly inhibit Sox4 expression in a process that facilitates AML development (Zhang, Cancer Cell 2013). Thus, we hypothesized that increased expression of Sox4 could possibly functionally lead to the hyper-sensitivity state.

Regarding the frequency of Sox4 alterations in malignancy, Sox4 is found overexpressed in 107 (23%) of 462 unique studies in over 20 types of cancer (Moreno, Semin Cancer Bio 2020). For example, Sox4 overexpression was reported to synergize with multiple oncogenes in malignant transformation (CREB, CEBPA and PML-RARa), in many preclinical models and human cells. In hematopoiesis, Sox4 expression is augmented when its direct negative regulator CEBPA is mutated (Zhang, Cancer Cell 2013). This is consistent with the clinical data showing that TET2 mutations are frequent in CEBPA double allelic mutant AML patients, and those cases are associated with worse prognosis (Heyes, Nat Comm 2023; Grossman, Br J Hemat 2013; Konstandin, Blood Adv 2018).

We also added a sentence to the discussion to better highlight this concept:

Increased Sox4 activity is reported to synergize with oncogenes during the process of leukemic transformation⁶²⁻⁶⁵; its expression peaks in primitive cells and gradually decreases along myeloid differentiation³⁶. Moreover, Sox4 expression is increased when its direct negative

regulator CEBPA is mutated. Clinical data show that TET2 mutations are frequent in CEBPA double allelic mutant AML patients, and those cases are associated with worse prognosis⁸⁶.

We agree with the Reviewer that a definition of whether alteration in Sox4 levels shows an additive vs. cooperative mechanism of action is of importance, but technically challenging due to the intrinsic limitations of our in vitro clonal model (such as the very fast and timed differentiation kinetics of clones both in vitro and in vivo).

While it would be of relevance to investigate whether increased activity of other correlated TFs can be a driver of hypersensitization, we feel it is out of scope for this work. Overall our findings validate the concept that heterogeneity at the level of the chromatin can be correlated with a cell state of functional hypersensitization to Tet2 mutation. To strengthen this concept we also amended the text and added new plots to Figure S6D (former Figure S4D) as following:

By analyzing ATAC-seq tracks at the proximal promoter region of several transcription factors positively correlated with high Tet2 sensitivity, we confirmed higher accessibility in hyper-sensitized clones (Fig.S6D). These data collectively reinforce the notion that within the same differentiation state extensive epigenetic heterogeneity is observed, and we identified a permissive epigenetic state correlated with higher molecular perturbation upon Tet2 KO.

11. More details needed on how functional perturbation score is normalized and calculated.

We apologize for the lack of clarity.

We have generated a separate methods section which now describes how the functional perturbation score is calculated as follows:

Raw data outputs obtained from functional assays performed in WT and KO GMP clones (expression of Ly6G, expression of Cd115, expression of Cd11b, expression of IL6, ROS production, phagocytosis after exposure to E.Coli and S.Aureus particles) were first standardized in order to render them comparable using the R function scale. A distance matrix among the different clones was then calculated using the R dist function and the euclidean distance measurement. The functional perturbation score is then defined for each GMP clone as the distance between WT-KO paired clone pairs.

12. Does SOX4 controls the differentiation of Neutrophils? Is there any molecular mechanism on why SOX4 OE TET KO cells accumulate MPP4 cells and decreased mature differentiation?

The exact role of Sox4 in shaping myeloid differentiation is not well established in the literature. However, some studies indicate that Sox4 has a functional relevance in healthy and diseased settings. It is expressed throughout myeloid differentiation (see Figures below) and was also reported to be upregulated in early neutrophil-committed progenitors during emergency myelopoiesis (Ikeda, Cell Rep 2023), suggesting an active role in myelopoiesis.

Response to reviewers, Figure 1. Sox4 regulatory expression within hematopoietic hierarchy. Top panel: DORC and RNA values from our own single cell datasets. Bottom: RNA expression determined with BloodSpot (<https://servers.binf.ku.dk/bloodspot/>)

Regarding the link between Sox4 and differentiation state, previous work (Zhang et al., Cancer Cell 2013) has established a direct role for CEBPA in repressing Sox4. Disruption of this interaction was reported to counteract myeloid differentiation, promote cell proliferation and ultimately contribute to transformation in AML setting.

Moreover, from our own data (Figure 6) we show that Sox4 overexpression drives increased Notch signaling and the induction of a glycolytic state which might be further supporting the increased competitive advantage of HSPCs in vivo. Lastly, MPP4 are reported to be a highly plastic progenitor population; while normally mainly contributing to lymphoid output, in aged context they also provide substantial contribution to myeloid lineages (Young, J Exp Med 2016). Sox4 expression is physiologically expressed at low levels in MPP4 vs HSC fractions (Sommerkamp, Blood 2021). We thus speculate that MPP4 might be more sensitive to alterations in levels of Sox4.

13. What is the status of lymphoid cells in the mice transplanted with SOX4 OE TET2 KD cells, was there any changes in the percentage of the lymphoid?

We report in former Figure S5B (now Figure S7B) no change in the percentage of mature B cells in the BM of Sox4 OE Tet2 KO mice as compared to Tet2 KO alone. We speculate that the absence of an appreciable lymphoid phenotype is due to the mild overexpression levels of Sox4 as driven from the PGK promoter (Bovia et al, Blood 2003).

14. Mechanisms on how SOX4 can be activated or epigenetic priming in GMPs needs to be commented on.

We thank the Reviewer for raising this interesting discussion point. We speculate that differences in levels of Sox4 expression can potentially arise by clonal selection. For example, as also mentioned in response 10, Sox4 overexpression is reported in hematopoietic clones harboring inactivating mutations of its negative regulator CEBPA (Zhang, Cancer Cell 2013). This is a described mechanism which fosters clonal transformation. We added a sentence in the discussion to highlight this concept:

We identified Sox4 as a driver of hyper-sensitization state towards Tet2 inactivation. Increased Sox4 activity is reported to synergize with oncogenes during the process of leukemic transformation⁶²⁻⁶⁵; its expression peaks in primitive cells and gradually decreases along myeloid differentiation³⁶. Moreover, Sox4 expression is increased when its direct negative regulator CEBPA is mutated. Clinical data show that TET2 mutations are frequent in CEBPA double allelic mutant AML patients, and those cases are associated with worse prognosis⁸⁶. Our data are consistent with a model where Sox4 promotes the maintenance of an immature and hyperproliferative cell state, which increases repopulation advantage and likely the susceptibility of accumulation of secondary mutations. Importantly, this concept was also validated using a clonal tracing method in primary HSC.

15. The authors should strongly consider depositing the code in a public repository for reproducibility.

We certainly agree and have done so: https://github.com/buenrostrolab/GMP_analyses_code

16. Additional mice in the control group of Tet2 fl/fl should be considered for the initial experiment (Fig. 1). Since authors state there are changes in cell frequency, statistical analysis supporting this claim are needed to estimate the reproducibility across biological replicates.

We agree with the Reviewer that increasing the number of mice would allow a systematic evaluation of Tet2-mediated effects on cell distributions. Lineage skewings upon Tet2 KO were however already extensively studied in the literature (Moran-Crusio et al., Cell 2011, Izzo et al., Nat Gen 2020 among others).

We now provide additional clarifications to the language, and new analyses to show the variability across biological replicates as following:

i) We clarified in the text that the analysis in Figure 1F is aimed to qualitatively assess small distortions of GMP cell states but not lineage skewings.

Finally, we analyzed how *Tet2* mutant cells are localized across the differentiation trajectory. In accordance with results from previous RNA-seq studies³², we observed that *Tet2* knockout does not result in independent cell clusters (**Fig.S1G**), but rather results in the redistribution of GMP cells within selected neighborhoods (**Fig.1F**). This suggests that *Tet2* mutation perturbs the chromatin state of primitive hematopoietic progenitors, but effects on cell identity are minimal since no significant skewing in cell abundances across clusters was observed (**Fig.S1J-K**).

ii) We added new panels aimed at depicting the variability across replicates: new Figures S1G and S1J show cell distribution on UMAP coordinates for each single mouse and fraction of cells belonging to each of the clusters, respectively. We do observe as expected some degree of variability in cell localization across individual mice (Figure S1G), but overall we don't detect consistent skewings in the abundance of GMP populations (further clarified in the text. We do only detect some batch-related alterations in the differentiated populations, which do not influence our downstream molecular analyses of GMPs and are likely due to the sorting procedure.

17. The cluster name "Mono-biased GMP" in Fig 1B is assigned to two different clusters. What are the differences between these two? Were they pooled for downstream analysis?

We thank the Reviewer and regret not being more clear. ATAC cluster identities were defined by pairing the ATAC data with cognate RNA data, which was in turn manually annotated. Please see also response to point 20 on more details and new panels included to clarify this concept. We also now revised the Cluster names in Fig.1B to specify both the cluster number and the population annotation. Clusters considered for each of the analyses in Figure 1 and 5 are now indicated in the figure legends.

Cluster 7 and 13 were both annotated as Mono-biased GMPs as correctly pointed out by the Reviewer. As shown by the new Figure S1I, RNA scores for canonical monocytic-related markers (*Ccr2*, *Klf4*, *Csf1r*, *Ly86*) show very comparable expression across the two different clusters with high overlap between Mono-biased GMP clusters as defined by scRNAseq in Figure S1I.

In terms of chromatin heterogeneity, we calculated the top variable 20 TF motifs over-represented for GMP clusters 7 and 13 (as respect to all GMP ATAC clusters). Whereas both show top hits in monocytic-specific motifs such as *Irfs*, *Bcl11a* and *Spib*, they show some dissimilarities such as *Fos/Jun/Nfkb* in cluster 13, suggesting that they represent different subtypes of monocytic precursors.

Top variable TF motifs		
#	Cluster 13	Cluster 7
1	Irf8	Irf8
2	Bcl11a	Bcl11a
3	Spib	Spib
4	Irf2	Spi1
5	Irf1	Irf2
6	Stat2	Irf1
7	Spi1	Stat2
8	Stat1	Prdm1
9	Prdm1	Stat1
10	Fos	Spic
11	Fosb	Zkscan5
12	Fosl2	Ets2
13	Irf7	Irf7
14	Zkscan5	Irf9
15	Nfkb2	Irf4
16	Nfkb1	Elf3
17	Irf9	Elf5
18	Jund	Zfp189
19	Rela	Zfp184
20	Irf4	Ehf

Response to reviewers, Figure 2. Top variable TF motifs for GMP clusters 7 and 13 (Mono-biased GMP).

Importantly, we also made available an interactive application (<https://buenrostrolab.shinyapps.io/gmps/>) which allows users to explore ATAC and RNA single-cell datasets reported in Figure 1 and 5. This provides an easy tool to further interrogate our single-cell dataset and provides a reference for further epigenetic studies evaluating the early and committed myeloid differentiation hierarchy in mice.

18. Labeling of the top annotation in the heatmap of Fig. 1D is required as well as color encoding. Color scale for each feature should be shown.

We apologize for not including these important details. We have amended Figure 1D including color encodings (which correspond to GMP clusters from Fig.1B), and color scales for each feature shown.

19. The authors show the residuals between DORC and RNA expression for Klf4 in particular (Fig. S1E). However, these dynamics are not show for control and Tet2 ^{-/-} separately, which seems like a missed opportunity. Do the dynamics differ between genotypes? Can this be associated with the monocyte progenitor skewing?

As suggested, we added dynamics of chromatin residuals (Chromatin-RNA) divided by genotype using Klf4 and other representative genes involved in myeloid development (new Figure S2D). While Klf4 residuals are not altered in Tet2 KO samples, we detect significant alterations for other myeloid-related residuals such as Irf8, which point to the functional alterations in Tet2 KO myeloid lineages.

This new data underscores the concept that chromatin accessibility can serve as an early indicator of gene expression changes and can be preferentially utilized over transcriptomic

studies for detecting alterations in cell state and function, particularly in cases of subtle differences introduced by Tet2 KO.

It's worth noting that this concept is also supported by our clonal analyses in Figures 3 and 4, where we leverage chromatin features in GMP cells to predict functional alterations in differentiated myeloid progenies. Together, these findings strengthen our argument that chromatin features play a crucial role in understanding and predicting changes in cell state and function in the context of Tet2 deficiency.

Lastly, we also added a new DORC differential analysis performed across individual GMP clusters (Table S3, Fig.S3C) which provides a complementary genome-wide assessment to the differential peak analysis and points to alterations in several myeloid-related regulators.

20. The authors should consider adding orthogonal validation of the cell cluster identity using reference mapping to verify the manually defined cell cluster labels.

We thank the Reviewer for the feedback.

To further clarify how the ATAC cluster annotations were determined, we did the following:

i) we revised former Figure S1B (now Figure S1H) and its legend to specify that the ATAC cluster annotation was made by pairing with cognate RNA dataset. The correlation between RNA-based clustering (manually annotated) and ATAC clusters is shown

ii) We added a new panel (Figure 1I) representing the RNA score for curated hematopoietic makers utilized for population annotation. We utilized gene signatures curated from Paul et al., Cell 2015, Giladi et al. Nat Cell Bio 2018, Olsson et al. Nature 2016, which collectively present a comprehensive panel of RNA markers to faithfully annotate different stages of early myeloid differentiation.

21. Claims such as “we observed differential localization across different WT clones (FigS4C)” are not clear. What is the definition of “differential localization”? How was it calculated? What statistics were used? A clearer language or statistical support is required for these claims.

We apologize for the unclear language. We modified the text as following:

We thus sought to better characterize heterogeneity among SCF clones. We performed combined scRNAseq and ATACseq analysis using SHARE-seq⁴⁰ on a subset of SCF GMP clones characterized by different functional properties. Visualization of some key motifs and marker genes allowed us to identify regions characterized by different levels of activity of key differentiation regulators (Fig.4G). We then observed that the distribution of cells across different WT clones is not overlapping. but rather single clones accumulate in specific regions of the UMAP coordinates (Fig.4F, Fig.S6C). This result further validates that functional heterogeneity observed upon differentiation is preceded by extensive molecular priming at the level of progenitor cells.

A new method section describing the “cell localization” metrics was also added (please see response to point 25).

22. In Figure 2C, the x axis labels are not clear. What is observed and what is expected? Is this Tet2 KO – WT? A clearer label should be used, and details should be included in the figure

legend.

We apologize for the mis-label. As suggested we now amended Figure 2C including the correct label and details in the figure legend.

23. In some figures the axis labels are overlapping with the numbers indicating the values of the variable.

We apologize and have thoroughly checked the axis labels and amended them to improve readability.

24. In Figure 4A, it is not clear why the WT column is necessary since it has been normalized to 1 for all clones. Consider showing the dispersion plotting Tet2 KO to WT Euclidean distance directly.

We now amended Figure 4B,C (former 4A,B) according to the Reviewer's suggestion removing the WT bar. Also, we additionally separated clones according to the cell type (SCF vs GM-CSF, see also comment 11 from Reviewer 2).

25. In Figure 4F, the authors show "% of localization". This metric is very obscure and not explicitly detailed in the main text nor the methods section. It is hard to interpret the results without that information.

We apologize for this missing detail. We added a paragraph to the method section as follows:

Cell localization analysis

For visualization of sample distribution on UMAP coordinates (Figures 1F, 5G), the % cell neighborhood per cell that belongs to a specific sample was represented for Tet2KO and Tet2KO + Sox4OE cells (relative to wild type cells). For each cell, the k-nearest neighbors were considered (k=50) using the batch-corrected (harmony) principal components, and the fraction of neighborhood cells that came from a non-wild-type genotype was determined, and shown on the UMAP. All mice were utilized for this analysis.

Similar analysis was performed for Figures 4G and S6C, representing the % cell neighborhood per cell that belongs to a specific sample for each SCF-derived GMP clone (relative to all other clones).

26. Figure 5E: It is not clear the metric that is being represented here. Also, the color legend states "log FDR". Assuming this is $-\log_{10}(\text{FDR})$, obtaining a value of 300 indicates that the statistic is not well calibrated.

We thank the Reviewer for spotting these inaccuracies in Figure 5E. We amended the label and the figure legend to clarify the metric plotted.

In this analysis we calculated the overlap between differential peaks in GMPs (Tet2 KO and Sox4OE Tet2 KO vs WT, respectively from Fig. 2A and 5D) and GMP chromatin signatures from Fig.1. Given the large number of peaks included in these analyses, the application of Fisher's test results in low FDR scores. We would like to clarify that such high values are not uncommon in genomics studies, particularly when dealing with large datasets and highly significant results. It does not necessarily indicate a calibration issue, but rather the strength of the evidence for the observed associations. While we acknowledge that alternative statistical tests could be

considered, we believe that the choice of statistical metrics employed does not significantly impact the biological conclusions drawn from the data.

27. The reasoning for performing metabolomic analysis is not entirely clear, and seems somewhat disconnect from the rest of the work.

We now modified the text of Figure 6 as follows to highlight the reasoning behind performing metabolic analysis in Sox4 OE cells:

To gain more understanding of the molecular bases for how the Tet2- hypersensitive cell state induced by Sox4 can alter cell phenotype, we performed untargeted metabolomic analyses (Fig. 6E). Notably, we observed greater glycolysis in hypersensitive cells. Previous studies have shown that Sox4 and Cdk8 expression correlate with changes in cell metabolism, in particular promotion of a glycolytic state^{79,80}.

Overall, the metabolic data provide further insight into how the hypersensitive cells are poised for hyperproliferation in CH.

Reviewer #2 (Remarks to the Author):

Clonal hematopoiesis (CH) is a widely studied age-related phenomenon associated with increased overall mortality due to elevated risks for haematologic malignancies and cardiovascular diseases. One of the most frequently mutated genes in CH is TET2. A poorly understood aspect of CH is its incomplete penetrance, with most patients remaining ostensibly disease-free, suggesting additional alterations are required for ostensible pathology to manifest itself. The manuscript by Schirolli et al. investigate in a knockout Tet2 mouse model how different cell states/cell of origins are functionally impacted by the loss of Tet2. The authors conduct single cell ATAC-seq and RNA-seq profiling of the murine bone marrow progenitor compartment to identify genomic alterations upon the loss of Tet2. An in vitro immortalized GMP model is then used to study genomic and functional differences upon Tet2 loss in clonal pairs of WT and KO clones, in conjunction with various functional analyses (e.g., phagocytosis, ROS production). The authors further identify Sox4 as an interesting candidate to cooperatively be responsible for the effects of Tet2 loss (e.g., Sox4 enhances the epigenetic dysregulation observed after Tet2 KO).

The manuscript is well-written, of overall very good quality, and of direct relevance to our understanding of CH. However, several aspects appear only touched upon in a rather superficial manner, leaving various loose ends. Moreover, the comprehension and readability of the work could be improved further by providing important details as outlined below. A more direct discussion of the relevance of their findings to CH would further be very valuable. This reviewer has the following major comments:

We thank the Reviewer for appreciating the quality of our work. We have extensively modified our text and figures to improve the readability and clarity of presentation. We also amended the discussion to more directly address the relevance of findings (and limitations) as following:

Our work extends this concept to pre-malignant and pre-mutation states and specifically defines that the epigenetic features of the clone before a mutation occurs can determine the

phenotype acquired. For CH, this finding can provide insight into the incomplete penetrance for disease development observed in patients. Although the specific CH driver mutation will likely determine the specific mechanisms of transformation, heterogeneity at the level of the clone of origin (within HSC and early progenitor compartments) could affect long-term expansion or facilitate further clonal selection and acquisition of secondary mutations. Further, we identified Sox4 activity as facilitating Tet2-related molecular alterations that foster clonal outgrowth in vivo. Our results validate the hypothesis that epigenetic features can predispose mutant hematopoietic clones for transformation and underscore the importance of pursuing prospective clonal tracking studies for the identification of such risk states for CH malignant evolution.

Limitations of the study. A potential limitation of our study is the use of an in vitro clonal system for the identification of Tet2-related functions. This model may be suboptimal mainly because i) cells are conditionally immortalized using HoxB8, which could influence chromatin accessibility and alter regions bound by Tet2 and ii) cells are arrested at the GMP state, therefore missing information regarding more primitive progenitors relevant for disease progression. Complementary in vivo studies using clonally tracked HSC-based models such as recently described²⁷ could allow uncovering of other molecular players that determine clonal advantage in CH models. Moreover, a clear distinction between phenotypic effects driven by differentiation stages or heterogeneous epigenetic states within the same cell type can be difficult to determine as these two cellular properties are closely related.

1. Another depiction of the skew in lineage trajectories in Fig. 1F should be provided. Is this skew consistently reproduced across the different mice that were used? Potentially bar plots resolved by cell state would facilitate such a quantification.

As suggested by the reviewer, we now provide additional clarifications to this point:

i) ii) We clarified in the text that the analysis in Figure 1F is aimed to qualitatively assess small distortions of GMP cell states but not lineage skewings as following:

Finally, we analyzed how Tet2 mutant cells are localized across the differentiation trajectory. In accordance with results from previous RNA-seq studies³², we observed that Tet2 knockout does not result in independent cell clusters (Fig.S1G), but rather results in the redistribution of GMP cells within selected neighborhoods (Fig.1F). This suggests that Tet2 mutation perturbs the chromatin state of primitive hematopoietic progenitors, but effects on cell identity are minimal since no significant skewing in cell abundances across clusters was observed (Fig.S1J-K).

ii) We added new panels aimed at depicting the variability across replicates: new Figures S1G and S1J show cell distribution on UMAP coordinates for each single mouse and fraction of cells belonging to each of the clusters, respectively. We do observe as expected some degree of variability in cell localization across individual mice (Figure S1G), but overall we don't detect consistent skewings in the abundance of GMP populations (further clarified in the text above). We do only detect some batch-related alterations in the differentiated populations, which do not influence our downstream molecular analyses of GMPs and are likely due to the sorting procedure.

iii) We detailed the method section to describe the mathematical calculation used and changed the labels on the figure to improve readability of the figure.

2. What's the degree of splenomegaly? Fig. S2B only provides a binary and no quantitative readout.

We amended Figure S2 and added a quantification of splenomegaly by representing spleen weight as percentage of body weight (please see new Figure S2C). When inducing tet2 KO by PI:PC treatment, we observe a ~2 fold increase in spleen size relative to body weight in a fraction of animals. This increase is largely consistent with the reported degree of splenomegaly for this mouse model (Moran-Crusio, Cancer Cell 2011).

3. The authors describe a variety of findings but show no matching data, which needs to be corrected throughout. Examples include:

- "Efficient Tet2 deletion was confirmed at the genomic level by ddPCR in >95% of circulating cells."
- "We also consistently observed a decrease in Ctf motifs, which are sensitive to DNA methylation."

We thank the Reviewer for highlighting some missing data.

A new Figure was added (Figure S1A), showing efficiency of Cre-mediated recombination by ddPCR.

Regarding the comment on Ctf motifs, we added the corresponding figure reference (Fig.2C) in the text.

4. Along these lines, basic quality control data should be shown for all single cell genomic datasets in the study. How many cells were profiled, pass QC, number of cells per mice, etc. Is data quality comparable, or is the signal driven by or more pronounced in 1-2 mice, given biological heterogeneity etc?

We apologize for the lack of clarity. As suggested, we now provide the following:

i) A new Table S1 which contains general information about the number of profiled cells passing QCs (according to the criteria defined in the methods section) for single cell RNA and ATAC seq datasets for each sample analyzed. Further information is also included in the GEO records where the raw data are deposited.

ii) Two new panels (Figure S1G, S1J) which show cell distribution on UMAP coordinates for each single mouse and fraction of cells belonging to each of the clusters.

Please note that in these tables the different experimental batches are also indicated.

5. Regarding the analysis in Fig. 2A and the following, which GMP populations were compared? As four GMP subpopulations were identified, with two belonging to a GMP-Mono state, this analysis should be conducted in a cluster-specific manner, in particular given some potential skew in the frequency of these populations as suggested in Fig. 1F.

In order to address this concern, we did the following:

i) we first generated two new panels (Figure S1J-I), which collectively show the distribution of cells across the different clusters. Apart from some batch-effect related changes in the

proportion of cells belonging to differentiated clusters, we do not detect significant skewing in the GMP-specific clusters (1,4,7,10,13).

We further clarified this concept in the text and highlighted that the UMAP from Figure 1F suggests small changes in cell states rather than a redistribution of cells across clusters:

Finally, we analyzed how Tet2 mutant cells are localized across the differentiation trajectory. In accordance with results from previous RNA-seq studies³², we observed that Tet2 knockout does not result in independent cell clusters (Fig.S1G), but rather results in the redistribution of GMP cells within selected neighborhoods (Fig.1F). This suggests that Tet2 mutation perturbs the chromatin state of primitive hematopoietic progenitors, but effects on cell identity are minimal since no significant skewing in cell abundances across clusters was observed (Fig.S1J-K).

ii) We clarified in the text that the differential analysis performed on pooled GMP was normalized by cluster identity as a covariate (as indicated in the corresponding methods section). We further highlighted in the legends of Figure 1 and 5 and methods section which populations were considered for the differential analysis.

iii) Although we agree with the Reviewer that cluster-specific comparison would provide useful insights, we provide a new panel (Figure S3A) where we model the scenario when differential analysis is performed by individual cluster. By repeating the differential analysis in bulk GMPs or using only smaller fractions of cells in the dataset, we show that our differential analysis is underpowered when analyzing small cell numbers. We believe that increasing the sensitivity of detection of Tet2 differential peaks is really valuable in this context.

iv) Although a differential analysis by single cluster is underpowered, we checked the directionality of differential peaks (calculated in the bulk GMP fraction) within individual clusters.

Response to reviewers, Figure 3. Directionality of differential peaks (calculated in the bulk GMP fraction) measured in individual GMP clusters.

As shown in the Figure above, overall directionality is largely maintained across individual clusters. This data reinforces the concept that our findings are not dependent on individual cluster distribution.

v) Since the differential analysis is underpowered, we now added a DORC differential analysis performed across individual GMP clusters, showing extensive alterations at the level of regulators of myeloid differentiation (new figure Fig.S3C, new Table 3).

6. Along these lines, this reviewer finds it surprising that differential up- and down-regulated peaks are so distinctly distributed across the chromatin landscapes of these cells (Fig. 2D). Do the authors have an explanation for this phenomenon? Are some of these differential changes already discernible in the respective later or earlier stages? Or phrased another way, are the differentially downregulated peaks a consequence of the previously upregulated peaks (resolving this relationship would go a long way in this reviewer's opinion)? Is there any correlation with Tet2 expression levels across the landscape (in the wildtype setting)? Can the authors show examples of differential chromatin peaks in track plots to give the reader a better intuition of these findings?

We apologize if Figure 2 text was lacking clarity regarding the type of differential analysis performed. We want to clarify that analysis from Figure 2D represents cumulative accessibility of GMPs differential peaks and not calculated across the entire dataset. Therefore the results can be collectively interpreted as alterations of GMP physiological differentiation. We also amended the figure label and legend to clarify which data are plotted in Fig.2D, as well as the text as following:

As a complementary approach, we assessed the differentiation stage where Tet2-differential chromatin regions calculated in GMPs are activated (Fig.2D). Peaks induced upon Tet2 KO showed high accessibility in primitive cells, whereas peaks repressed upon Tet2 KO showed highest accessibility upon progenitor commitment towards myeloid fates. These data are consistent with Tet2 being required for priming of GMPs towards both monocytic and granulocytic lineages.

As suggested by the Reviewer, we also add a new panel showing track plots of Tet2 differential peaks in GMPs (new Figure S3B) to further highlight that differential analysis is performed in GMP populations.

Tet2 is overall expressed throughout the hematopoietic differentiation hierarchy (see plot below, analysis performed using BloodSpot, <https://servers.binf.ku.dk/bloodspot/>). Tet2 expression is not detected with robustness in single-cell datasets including our own.

Response to reviewers, Figure 4. Tet2 expression within hematopoietic hierarchy.

However, Tet2 expression is not a full predictor of activity, since Tet2 does not contain a direct DNA binding domain but relies on interaction with TFs for exerting its demethylating action.

7. Why was no matching analysis to Fig. 2A conducted for the HSPC compartment? As in particular the GMP-stem signature extends to this compartment (Fig. 1E), and many of the differential peaks are also identified in HSPCs (Fig. 2D) it appears more than worthwhile to provide a comprehensive analysis across the trajectory of monocytic differentiation, be the major focus the GMP compartment.

We agree with the Reviewer that a full comparison of Tet2 effects across the myeloid differentiation hierarchy would be of interest. To answer to this concern, we did the following:

i) As fully stated in response to comment 5, our differential analysis is underpowered when analyzing small cell numbers. As shown in the new panel Figure S3A, we model the scenario when differential analysis is performed by individual cluster and highlight a strong dependence on numbers of input cells. This renders challenging to perform a quantitative comparison of Tet2 impact across myeloid differentiation trajectory. For example, by repeating the differential peaks analysis on a smaller cluster like the HSPC one, we did detect only 2 differential peaks.

As a qualitative measure, we checked the directionality of differential peaks (calculated in the bulk GMP fraction) within HSPC and mature myeloid clusters. As expected, we observe some levels of similarity between effects of Tet2 within these populations but also several differences which likely reflect different TF usage across myeloid differentiation hierarchy (Pundhir et al., Cell Reports 2018).

Response to reviewers. Figure 5. Directionality of differential peaks (calculated in the bulk GMP fraction) measured in individual HSPC and mature myeloid clusters.

ii) We further clarified in the text and figure legend that analysis from Figure 2D represents a measure of cumulative accessibility of GMP differential peaks and not of differential peaks calculated across the entire dataset. Please also see response to comment 6.

iii) Lastly, since the differential analysis is underpowered, we added a new Figure (Fig.S7E) depicting quantile-normalized DORC accessibility score for some representative genes relevant for differentiation, across the differentiation trajectory. This new data highlights some examples of epigenetic dysregulation in both Tet2 KO and Sox4 OE Tet2 KO conditions as respect to WT. See also response to point 5 for a more detailed description of the differential DORC analysis performed.

8. Regarding Fig. 2G, what does ROS production at baseline (-PMA) look like? Is there already a difference between WT and Tet2 KO? Also, the statement “Mature Tet2 KO myeloid cells from both lineages showed increased reactive oxygen species (ROS) production upon treatment with PMA (Fig.2G). This finding is consistent with a described role for Tet2 in repressing inflammatory gene expression.” needs to be clarified. How is ROS production here linked to alterations in gene expression?

We now added a new panel in Figure 2G (right panel) measuring ROS production at baseline in an independent cohort of WT and Tet2 KO mice. We don't see baseline alterations in ROS production, suggesting that Tet2-mediated functional alterations are increased only under inflammatory conditions. Importantly, these data are also in accordance with findings from Tet2 KO GMP clones reported in Figure 3 and S3 (see new panel 3D).

We also removed the confusing sentence “This finding is consistent with a described role for Tet2 in repressing inflammatory gene expression” in an attempt to streamline the text of Figure 1 and 2.

9. About Fig. 3B: How do WT and Tet2 KO clones cluster in 3B? Presumably, the pairs cluster

closely together, or are the alterations more obvious than in the prior data?

We now added a new panel in Figure 3B (right panel) showing clustering by genotype. Overall, we see genotype-specific clustering (as opposed to intermixed populations in the data of Figure 1), highlighting the advantage of using clone-based systems. We indeed observe that Tet2 KO extensively alters the chromatin accessibility profile of GMP clones, and these epigenetic changes are then further reflected in functional alterations.

10. Data in Fig. S3D-G should be summarized, showing mean and standard deviations. The 29 pairs of bar plots should be kept in the supplement, but these alone are challenging to digest. Can the authors show chromatin track plots for the differential loci, e.g., IL6, Cd115, Ly6G.

We agree with the Reviewer that understanding the effects of Tet2 KO on GMP clones is not easy with the current data. As suggested, we now added a new panel (Figure 3D) showing the impact of Tet2 KO as a barplot (respect to WT control clones). We believe it is now easier to appreciate the functional alteration observed in KO cells (myeloid differentiation markers, phagocytosis, production of ROS and IL6).

As suggested, we also now provide track plots for representative WT GMP clones for differential loci identified by functional assays in differentiated cells (markers of differentiation and inflammation) in a new panel (Figure S5E). This novel data shows correlation between chromatin status at the level of GMPs and downstream functional properties of effector cells. See also response 8 to Reviewer 1.

11. About the subsequently introduced functional perturbation score. Can the authors provide more details on the methods on how exactly this has been calculated? What's the major driver of that score? Also please highlight the GM-CSF clones in Fig. 4B, or were these not transplanted? Does the *in vivo* output correlate with the functional perturbation score?

We thank the Reviewer for the constructive feedback. We overall revised Figure 4 to improve the clarity on how the calculated perturbation score relates to individual phenotypic changes and to the SCF/GM-CSF identities. We added a new panel (Figure 4A), which shows the fold perturbation in each measured *in vitro* functional property for each individual clone pair (Tet2 KO-WT). We also revised former Figure 4A,B (new Figures 4B,C) to show functional scores divided by SCF/GM-CSF identities. To further improve readability and cross-analysis comparison between Figure 3 and 4 we added clone ids to each of the presented heatmaps.

Overall, we believe the new data presentation better highlight the following concepts:

- i) GM-CSF derived clones are much less perturbed in their behavior as compared to SCF clones, suggesting that the differentiation state of cells plays a role in determining the extent of Tet2-mediated phenotypic changes.
- ii) Within SCF (less differentiated) clones, we can observe a high degree of functional heterogeneity in the response to Tet2 KO. In particular the ROS production, IL6 production and phagocytosis are the assays where we see more variability across clones (and thus have a higher impact in driving the overall functional perturbation score).

We also generated a separate methods section which now describes how the functional perturbation score is calculated:

Functional perturbation score

Raw data outputs from obtained from functional assays performed in WT and KO GMP clones (expression of Ly6G, expression of Cd115, expression of Cd11b, expression of IL6, ROS production, phagocytosis after exposure to E.Coli and S.Aureus particles) were first standardized in order to render them comparable using the R function scale. A distance matrix among the different clones was then calculated using the R dist function and the euclidean distance measure. Functional perturbation score is then defined for each GMP clone as the distance between WT-KO paired clone pairs.

Finally, GM-CSF clones do not give rise to an appreciable graft upon in vivo transplantation, possibly due to the terminal differentiation state at which they are arrested (Wang, Nat Meth 2006) and thus were not assayed by in vivo transplant. Overall, we observe a mild correlation between the in vivo output of the clones and the functional perturbation assayed in vitro (see plot below), suggesting that these two scores capture complementary cellular properties. Of note, we did not perform in vivo transplantation for all the clones assayed.

Response to reviewers, Figure 6. Correlation between in vitro perturbation score and In vivo output for a number of SCF clones.

12. The authors nominate Sox4 as an interesting gene candidate from their GMP clone data. Do the authors also see any indication of Sox4 activity/alteration in their Fig. 1-2 data? Please show flow plots for the Fig. 4H data (presumably flow cytometry was used here), as differences in CD117 expression appear rather subtle and the bar plots do not account for the distribution of the signal. Can the authors comment on how Sox4 and Tet2 may cooperate at the chromatin level? Do they bind to distinct or the same target genes? Is the Sox4 motif and as such target gene regulatory elements known?

We thank the Reviewer for raising this interesting point. We do not see alterations in Sox4 activity or gene expression in Tet2 KO mice. Based on our hypothesis, we don't expect Tet2 to influence levels of Sox4, but rather posit that small differences in endogenous levels of Sox4 prime cells for higher functional sensitivity to the effect of the mutation.

Future studies will be needed to define the mechanism of cooperation between Sox4 and Tet2 KO. Based on our data, we speculate the following:

i) Sox4 acts independently from Tet2 as a transcriptional activator, according to its described function. It is indeed reported to directly bind DNA through a conserved high-mobility group DNA binding domain and directly regulate stemness, differentiation and progenitor development. Our data in progenitor cells indicate indeed that Sox4 is directly activating transformation-promoting pathways such as Notch, Fgf and PI3k.

ii) Sox4 also partially acts by potentiating Tet2-mediated epigenetic dysregulation, as evidenced by our scATACseq dataset. We do not know the precise mechanism of action, but Sox4 could either directly or indirectly affect the availability of TFs directly interacting with Tet2.

As requested, we included two new panels to Figure 4J (former Figure 4H) to exemplify the data distribution from the flow cytometry analysis. We now represent the changes in mean fluorescence intensity of CD117 and CD11b as fold change compared to day 0 to account for clone-specific alterations. Although the changes in MFI are modest, the fast and tightly timed kinetics of in vitro differentiation of HOXB8 immortalized GMP cells (see also Sykes et al, Cell 2016) suggest that modest Sox4-induced alterations represent meaningful differences in the differentiation process.

12. Please show an annotated UMAP of the dscATAC-seq data discussed in Fig. 5. While a UMAP is shown in 5F, its unclear whether this relates to data in Fig. 1A, or whether the new data was projected onto the same landscape, etc. As noted above, basic QC metrics should be shown, alongside how consistent the results are across the mice/replicates used. Analogous to the previous point 5, which GMP (sub)population(s) was analyzed?

We apologize for the lack of clarity. We now amended the main text and figure legends to underline that all the data plotted in Figure 1 and Figure 5 were projected on the same UMAP coordinates. As requested, we also included two new Figures (Fig S1G and Fig S1J) showing the distribution of cells within clusters and for each single mouse. We also added a Table (Table S3) reporting the QC metrics for the sc ATAC seq datasets. Finally, we amended the text and figure legends to clarify which GMP populations have been included in the differential analysis.

13. The authors should add a discussion about how directly their findings related to the CH in a human setting. One would expect that TET2 mutations will ultimately arise only in a self-renewing population, presumably HSCs. Would the authors therefore suggest that subtle differences between HSCs, predetermine long-term consequences, or that such a self-renewing population may further diversify and then additional subclones to become more susceptible to developing pathology? In particular, as the CH population may evolve over many years if not decades.

We thank the Reviewer for raising this interesting perspective. We believe that states of the cell of origin can contribute to both predetermine long-term consequences or predispose to further clonal selection, although only extensive in vivo clonal tracking studies will definitively answer the question. As suggested, we have therefore added the following sentence to the discussion to better contextualize our work with human findings :

Our work extends this concept to pre-malignant and pre-mutation states and specifically defines that the epigenetic features of the clone before a mutation occurs can determine the

phenotype acquired. For CH, this finding can provide insight into the incomplete penetrance for disease development observed in patients. Although the specific CH driver mutation will likely determine the specific mechanisms of expansion and transformation, heterogeneity at the level of the clone of origin (within HSC and early progenitor compartments) could affect long-term expansion or facilitate further clonal selection and acquisition of secondary mutations.

14. The GEO data is not accessible. A reviewer access token should be provided.

We apologize for the oversight and now provide a reviewer access token for the GEO data: spgpeksolripbov

Minor comments

1. Why did the number of GM-SCF (4) and SCF (24) derived clones used for analysis differ?

We utilized a high number of SCF clones to study how epigenetic heterogeneity within the same clonal state affects functional response after induction of Tet2 inactivating mutation (comparison within SCF clones). We believe this is the most relevant and interesting question, because it could provide some mechanistic insight on how clonal heterogeneity could explain the phenomenon of incomplete penetrance seen in humans.

GM-CSF clones were selected to answer a different biologic question: how cells arrested at different differentiation states respond to the same mutation (comparison between SCF and GM-CSF). Although this is an interesting question, it doesn't represent the main focus of our work, and for this reason we limited the number of clones utilized.

2. Fig. S3C: can the authors show pair-wise flow plots of the different markers? Based on the histograms alone, the phenotypic differences in marker expression are difficult to compare.

We feel that it is important to keep Figure S3C represented as an histogram in order to show comparative marker expression between SCF and GMCSF clones. In order to facilitate comprehension of the gating strategy utilized for the in vitro analyses from Figure 3 and Figure S3, as requested we also now provide an additional new Figure S3D which shows marker expression in a pairwise manner for a representative SCF clone (before and after differentiation).

3. Fig. S3D-G: please indicate the statistical tests used.

We thank the Reviewer for highlighting the missing description of statistical tests in figure S3. The figure legend is now amended and corrections are noted in red throughout the text.

4. Revise the use/introduction of abbreviations (should be introduced when used first, e.g., dsc, OE, pl:pC)

We thank the Reviewer for the note and spelled correctly all the abbreviations utilized in the text. Corrections are noted in red throughout the text.

5. Details in the materials and methods need to be revised for consistency (uM -> μM, C -> °C, 105 -> 10⁵/10E5, % -> f %(v/v), agitated (rpm?) 2^Δ-DCt instead of 2^ΔCT, 30' vs 30 min), etc.

We apologize for the inaccuracies in the method section. We now amended the methods section as suggested by the Reviewer. Corrections are noted in red throughout the text.

Reviewer #3 (Remarks to the Author):

In this very complex manuscript, the authors comprehensively examined the effects of TET2 deletion in different hematopoietic stem and progenitor populations in a conditional TET2 KO mouse model. They describe the epigenetic landscape of TET2 mutated hematopoiesis in detail and provide evidence that the epigenetic state of the cell of origin also determines the phenotype of the cell when TET2 is mutated. The authors then go on to identify SOX4 as a driver of increased myeloid output in their system by using an elegant system of immortalised myeloid progenitor cells enabling single cell expansion and exact timing of differentiation by stimulation with either SCF or GM-CSF in vitro. TET2 KO cells showed distinct chromatin accessibility patterns depending on stimulus towards neutrophil or myeloid-monocyte differentiation.

The manuscript contains extensive datasets and the authors are to be commended for their thorough characterisation of the epigenetic effects of TET2 deletion on hematopoietic subsets, which provides a valuable reference for other groups. However, the results section (especially pertaining to Figures 1 and 2) are at times quite difficult to follow, especially since the description of the results in figure 2 includes many references to the current literature without necessarily following up on this data in the rest of the manuscript. I think the results section would benefit from streamlining to increase readability and clarity without extensive discussion of observations that do not play a role in the rest of the story. The introduction and discussion in contrast are very clear and well written.

We thank the Reviewer for this constructive comment. We extensively streamlined the text relative to Figure 1 and 2 and highlighted the changes in the revised manuscript.

Figure 1F it seems that TET2 KO cells are also skewed towards an MEP phenotype but this is not recapitulated in colony data in Fig S2C. Can the authors comment.

Figure 1F is generated using data comprising all the analyzed mice which originate from 2 separate independent experimental batches (batch 1: 2 WT and 2 Tet2 KO mice; batch 2: Tet2 KO and 2 Sox4 OE Tet2 KO mice). We now provide two new panels (Figure S1G, Figure S1J) showing the distribution of cells for each single mouse and within clusters, respectively. As evident from Figure S1J, the relative increase in MEP (cluster 6) likely represents a product of experimental batch effect, specifically from the sorting procedure (mice from batch 2 all showed increased MEP representation). In support of this, previous literature has established a skewing against erythroid commitment in Tet2 KO mice (and not towards).

We further clarified in the main text that the focus population of Figure 1F is within GMP cells and that this analysis is aimed to highlight small state changes in the GMP populations (independently from the subsequent differential analysis performed Figure 2).

Fig 2B peak distribution WT vs TET2 KO (in black) is barely readable.

We apologize for the lack of clarity in the graphic representation of Figure 2B. We now amended Figure 2B to improve readability of the labels.

Fig 2C shows repressed peaks in chromatin regions for transcription factors that are involved in regulation of monocyte commitment: this is perhaps surprising as in the clinic one of major phenotypes of TET2 mutation is usually monocytosis. Can the authors discuss.

We thank the Reviewer for raising this important point. As correctly pointed out, the analysis in GMP cells Figure 2C, 2D and 2E shows enrichment of repressed peaks involved in monocytic and partially also neutrophilic commitment in Tet2 KO samples. Since Tet2 does not contain any known DNA-binding domain, the protein is likely recruited to DNA through interaction with a range of different DNA-binding proteins such as TFs. The identity of the interacting TFs depends on the specific cellular context/differentiation lineage (elegant demonstration of this concept is reported by Sardina et al., Cell Stem Cell 2018).

Based on this premise, we think that our data, which describes differential peaks in GMP being enriched in chromatin regions for myeloid commitment, is consistent with Tet2 physiologic mechanism of action.

To strengthen this concept, our data are also consistent with DNA methylation data from primary human samples with TET2 mutations (Tulstrup et al. Nat Comm 2021). This work describes DNA hypermethylation at enhancer sites in mature myeloid cells in the absence of TET2, and similarly reports enrichment of TF motifs in repressed regions implicated in myeloid cell function (ELF2,ETS1, CEBPA, CEBPD, SPI1).

Furthermore, we want to clarify that our data don't indicate that myeloid development is blocked in Tet2 KO context, but rather that physiologic maturation is altered.

In support of this, analyses of progenitor distribution (novel Figure S1J) suggest similar abundances across GMP subtypes and mature cells between WT and Tet2 mutant cells.

Our data are consistent with recent work from Huerga Encabo, Cell Stem Cell 2023. This work indeed highlights by transcriptomic, epigenomic and functional studies the accumulation of mature neutrophils blocked in some of their terminal effector functions in TET2 KO context but does not report a defect in reaching terminal differentiation.

Fig 2G recently published work has shown that TET2 mutated neutrophils differentiated from human HSPC in a NSG transplant model have defective phagocytosis (Huerga Encase et al, Cell Stem Cell 2023). So this observation is relevant, but not novel. This work should be cited in this context.

We appreciate the reviewer pointing this out and have amended the text referring to Figure 2G to include the reference from Huerga Encabo et al., Cell Stem Cell 2023 among the literature describing a defective role of Tet2 KO myeloid cells.

While the mouse model nicely recapitulates the myeloid bias in TET2 mutated CH, previous work (Buscarlet 2018) has shown that in human CH TET2 arises in a later cell of origin than for

example DNMT3A, i.e. committed HSC vs a multipotent HSC. How does this compare to the mouse data here. Could the occurrence of TET2 mutations lower in the human hematopoietic hierarchy explain the GMP heterogeneity seen in the mouse model?

We thank the Reviewer for raising this important point. Work from Buscarlet et al., 2018 analyzed variant allele frequencies in individuals with single *DNMT3A* or *TET2* mutations across multiple lineages. This work reported that *TET2* mutant cells were markedly enriched in the myeloid lineage, in absence of multilineage involvement in lymphoid lineage (using a variant allele frequency of 2% as a threshold to define presence or absence of mutations). While this data might be interpreted as the cell of origin of *TET2* mutations occurring in more committed progenitors, we think the alternative explanation the authors provide in the manuscript better supports the rest of the literature and our own findings: *TET2* mutations could occur directly in a multipotent HSC, but cooperation with other factors including epigenetic modifications may influence lineage proliferation bias.

Data obtained from mouse models supports this latter concept. Indeed, no ectopic self renewal has been reported in *Tet2* mutant early progenitors (Ostrander, Stem Cell Reports 2020) or GMPs (Shih, Cancer Cell 2015), suggesting that the cell of origin needs to be endowed with long term self renewal to propagate over years. Moreover, Izzo et al., Nat Genet 2020 reports that *Tet2* KO mice present with an expansion of the early HSC compartment, and pinpointed the observed myeloid skewing to an alteration in lineage priming due to hypermethylation of TF motifs involved in erythroid differentiation. This phenomenon can contribute to the myeloid skewing observed in *TET2* mutant patients. Similarly, recent work from Huerga Hencabo, Cell Stem Cell 2023 reports mechanistic insights on increased myeloproliferation within the mature granulocytic lineage in *TET2* KO cells.

In our manuscript, we provide evidence of clonal heterogeneity shaping the phenotypic effect of *Tet2* mutation, and identify *Sox4* as a cofactor. We validate this mechanism both in HSC-based models and GMPs, suggesting that increased proliferation is shared across different early differentiation stages and supporting the concept that phenotypic effects of myeloproliferation can be realized by expansions of HSC or downstream progenitors.

REVIEWER COMMENTS

Reviewer #1 (Remarks to the Author):

The authors have addressed all of my questions and concerns.

Reviewer #2 (Remarks to the Author):

The authors satisfactorily responded to the reviewer's comments and provided a significantly improved manuscript. However, some additional details should be addressed:

1. Only 2/6 mice show signs of splenomegaly. The wording of the manuscript should be adjusted to accurately reflect this.
2. New Figure S5E only shows chromatin track plots for WT but no TET2 KO clones, which should be added for completion.
3. Based on the response to the previous reviewer comment 6: This reviewer understands the challenges related to robustly detecting the expression of genes such as Tet2. For completion, Tet2 expression should nevertheless be shown as suggested, while commenting on challenges related to sensitivity and expression not reflecting activity.

Reviewer #4 (Remarks to the Author):

The authors satisfactorily answered to all Reviewer 3's concerns.

Minor comment: the lettering of the different panels of figure 4 is messed up in the figure legend.

Cell of origin epigenetic priming determines susceptibility to *Tet2* mutation

We thank the Reviewers for the positive re-evaluation of our manuscript. We have further amended the manuscript to fulfill the remaining comments and now hope our work can be now considered suitable for publication in Nature Communications. Please find a detailed point-by-point discussion below.

Reviewer #1 (Remarks to the Author):

The authors have addressed all of my questions and concerns.

Reviewer #2 (Remarks to the Author):

The authors satisfactorily responded to the reviewer's comments and provided a significantly improved manuscript. However, some additional details should be addressed:

1. Only 2/6 mice show signs of splenomegaly. The wording of the manuscript should be adjusted to accurately reflect this.

We amended the text to reflect this detail as following:

*Upon *Tet2* deletion, phenotypic analysis highlighted expansion of myeloid cells in the periphery, splenomegaly in a fraction of mice, and accumulation of primitive HSC and myeloid progenitors in the BM, consistent with previous reports for this model (Fig.S1B-E)³⁰.*

2. New Figure S5E only shows chromatin track plots for WT but no TET2 KO clones, which should be added for completion.

As suggested by the Reviewer, we now added to Figure S5E the chromatin track plots for the *Tet2* KO clones in addition to WT ones.

3. Based on the response to the previous reviewer comment 6: This reviewer understands the challenges related to robustly detecting the expression of genes such as *Tet2*. For completion, *Tet2* expression should nevertheless be shown as suggested, while commenting on challenges related to sensitivity and expression not reflecting activity.

As suggested by the Reviewer, we added a new supplementary panel (Fig.S3D) representing expression of *Tet2* measured in WT mice within our single cell dataset and complemented the main text and the discussion as following:

*(main text) Since our epigenetic analysis indicates extensive chromatin remodeling, we analyzed the downstream alterations at the level of gene expression. *Tet2* is expressed throughout the hematopoietic differentiation hierarchy, although poorly detected in our scRNAseq dataset (Fig.S3D). Enrichment analysis comparing *Tet2* KO and WT cells was conducted for cell subsets covering the key stages of myeloid differentiation (Fig. 2F).*

(discussion) Furthermore, our data highlight that individual cell types can be affected by Tet2 differently. *Although widely expressed*, Tet2 interaction with DNA relies on the interaction with chromatin modifiers and TFs, due to the lack of a direct DNA binding domain. This interaction enables a cell type and chromatin context-specific activity⁸⁹.

Finally, we also further implemented our interactive application which includes ATAC and RNA single-cell datasets (<https://buenrostrolab.shinyapps.io/gmps/>), adding the capability to filter samples by genotype. This new functionality will aid a more thorough interrogation of our single-cell dataset.

Reviewer #4 (Remarks to the Author):

The authors satisfactorily answered to all Reviewer 3's concerns.

Minor comment: the lettering of the different panels of figure 4 is messed up in the figure legend.

We apologize for the oversight and corrected the lettering of Figure Legend 4 to accurately mirror the corresponding figure.

Reviewers' Comments:

Reviewer #2:

Remarks to the Author:

The authors have addressed all of the reviewer's comments.